# TAPBPR promotes antigen loading on MHC-I molecules using a peptide trap

Andrew C. McShan[1,5], Christine A. Devlin [2,5], Giora I. Morozov[1], Sarah A. Overall[3], Danai Moschidi[3], Neha Akella[2], Erik Procko [2✉] & Nikolaos G. Sgourakis [1,4✉]

Chaperones Tapasin and TAP-binding protein related (TAPBPR) perform the important functions of stabilizing nascent MHC-I molecules (chaperoning) and selecting high-affinity peptides in the MHC-I groove (editing). While X-ray and cryo-EM snapshots of MHC-I in complex with TAPBPR and Tapasin, respectively, have provided important insights into the peptide-deficient MHC-I groove structure, the molecular mechanism through which these chaperones influence the selection of specific amino acid sequences remains incompletely characterized. Based on structural and functional data, a loop sequence of variable lengths has been proposed to stabilize empty MHC-I molecules through direct interactions with the floor of the groove. Using deep mutagenesis on two complementary expression systems, we find that important residues for the Tapasin/TAPBPR chaperoning activity are located on a large scaffolding surface, excluding the loop. Conversely, loop mutations influence TAPBPR interactions with properly conformed MHC-I molecules, relevant for peptide editing. Detailed biophysical characterization by solution NMR, ITC and FP-based assays shows that the loop hovers above the MHC-I groove to promote the capture of incoming peptides. Our results suggest that the longer loop of TAPBPR lowers the affinity requirements for peptide selection to facilitate peptide loading under conditions and subcellular compartments of reduced ligand concentration, and to prevent disassembly of high-affinity peptide-MHC-I complexes that are transiently interrogated by TAPBPR during editing.

[1] Center for Computational and Genomic Medicine, Department of Pathology and Laboratory Medicine, The Children's Hospital of Philadelphia, Philadelphia, PA, USA. [2] Department of Biochemistry and Cancer Center at Illinois, University of Illinois, Urbana, IL, USA. [3] Department of Chemistry and Biochemistry, University of California Santa Cruz, Santa Cruz, CA, USA. [4] Department of Biochemistry and Biophysics, Perelman School of Medicine, University of Pennsylvania, Philadelphia, PA, USA. [5] These authors contributed equally: Andrew C. McShan, Christine A. Devlin. ✉email: procko@illinois.edu; Nikolaos.Sgourakis@Pennmedicine.upenn.edu

Class I major histocompatibility complex (MHC-I) molecules bind a repertoire of endogenously processed peptide antigens and display them on the cell surface, thereby enabling CD8[+] cytotoxic T lymphocytes to surveil the cell proteome[1]. The disparate surface chemistries of peptide/MHC-I (pMHC-I) antigens encompassing self, pathogen, or tumor-derived peptides allows T lymphocytes to detect aberrant protein expression and mount a response toward infected or tumor cells[2]. Assembly of pMHC-I molecules occurs in the endoplasmic reticulum (ER), where the association between a peptide, the polymorphic heavy chain (human leukocyte antigen, HLA), and the invariant light chain ($\beta_2$-microglobulin, $\beta_2$m) is facilitated by dedicated molecular chaperones, the ER-restricted tapasin of the peptide-loading complex (PLC) and the PLC-independent TAPBPR (TAP-binding protein related)[3,4]. In addition to chaperoning MHC-I, tapasin and TAPBPR function as catalytic enhancers of peptide association and dissociation within the MHC-I groove (peptide exchange) and participate in peptide editing and quality control of assembled pMHC-I molecules[5–8]. Additional quality control is achieved by the association of TAPBPR with UDP-glucose:glycoprotein glucosyltransferase (UGGT1), which promotes reglycosylation of empty or suboptimally loaded MHC-I

towards their recycling to the PLC[9,10]. Collectively, these processes ensure the correct folding, trafficking and prolonged cell surface display of pMHC-I antigens[11]. Deregulation of chaperone function has been observed during disease progression in the context of autoimmunity, cancer, and pathogen infection[12–15].

Two crystal structures of TAPBPR-bound MHC-I molecules have illuminated aspects of the chaperoning and peptide editing functions[16,17]. TAPBPR has fused β-sandwich and immunoglobulin-like type 5 (IgV) folds forming a bilobed N-terminal domain with a large concave surface that cradles the underside of the MHC-I $\alpha_1/\alpha_2$ "platform", beneath the $\alpha_2$ domain. The peptide-binding groove of the MHC-I is held in an open conformation at the F pocket, where the antigenic peptide C-terminus would normally reside, with smaller distortions extending to the A and B pockets. These groove perturbations promote the release of low-affinity peptides. The C-terminal IgC domain of TAPBPR makes additional contacts to the MHC-I $\alpha_3$/$\beta_2$m domain junction, giving the appearance of a long scaffold supporting the different structural elements of the MHC-I. In one of the TAPBPR crystal structures[17], a protruding loop (residues G24-R36) of TAPBPR (Fig. 1a) was modeled in an α-helical conformation extending from the luminal tip of the

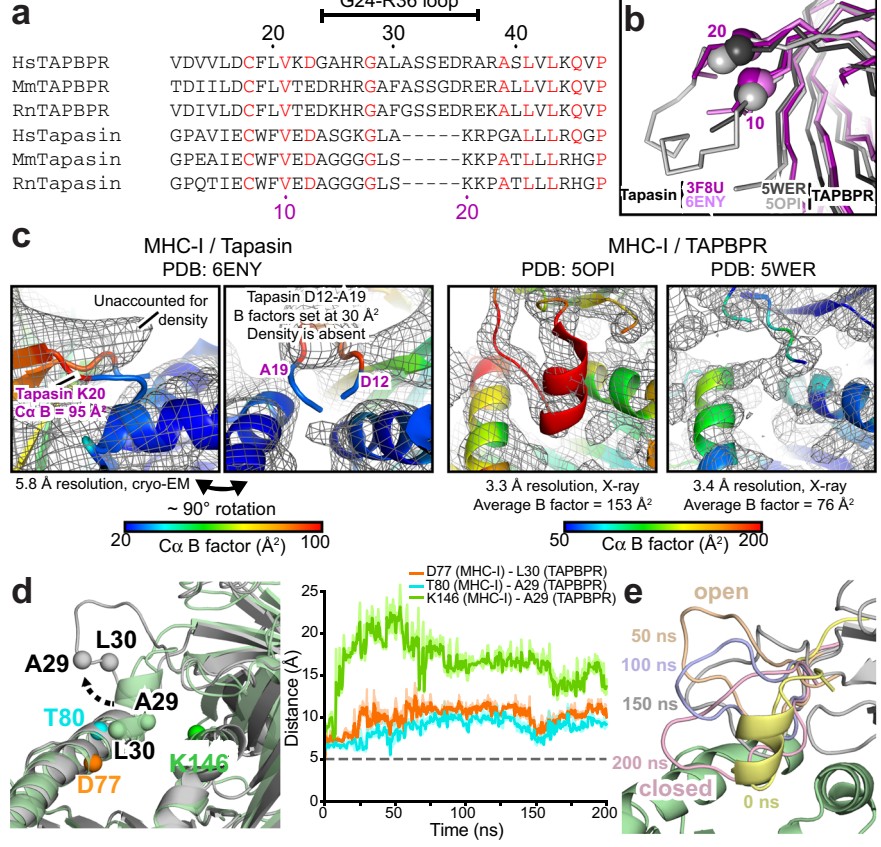

**Fig. 1 Conformational plasticity of the TAPBPR G24-R36 loop. a** Sequence alignment between TAPBPR and tapasin from *Homo sapiens* (Hs, human), *Mus musculus* (Mm, mouse), and *Rattus norvegicus* (Rn, rat) highlighting differences in the TAPBPR G24-R36 loop region. Black and purple numbering reference TAPBPR and tapasin, respectively. Red residues are conserved. **b** Overlay of X-ray structures of tapasin (PDB IDs 3F8U and 6ENY) and TAPBPR (PDB IDs 5WER and 5OPI). **c** (Left) Cryo-EM density map (gray mesh, 5.0 σ) plotted on the cartoon of MHC-I in complex with tapasin (PDB ID 6ENY). (Right) $2F_o$-$F_c$ electron density maps (gray mesh, 1.0 σ) plotted on the cartoon of MHC-I in complex with TAPBPR (PDB ID 5OPI and 5WER). The cartoons are colored by Cα B factors. **d** (Left) Before (green) and during (gray) snapshots of peptide-deficient HLA-A*02:01/h$\beta_2$m/TAPBPR complex from all-atom MD simulations. The Cα atoms used for distance measurements are shown as spheres. (Right) Intermolecular Cα-Cα distances measured between HLA-A*02:01 groove (D77, T80, K146) and TAPBPR G24-R36 loop (A29, L30) residues over the course of the simulation. The dotted line represents Cα-Cα distances at the start of the simulation. **e** The range of conformations of the TAPBPR G24-R36 loop captured at different times during the MD simulation. The open (wheat, 50 ns) and closed (pink, 200 ns) states of the TAPBPR G24-R36 loop are oriented away from and covering the HLA-A*02:01 groove, respectively. The MHC-I groove (green) and TAPBPR N domain (gray) are shown as a static snapshot from the 0 ns MD time point for clarity.

N/IgV domain into the peptide-binding groove to occupy the F pocket (Fig. 1b, c-middle). The loop was hypothesized to "scoop" bound peptide out of the groove through steric competition. A simultaneous publication from the same group of the tapasin-dependent PLC at low resolution seemed to support this mechanism, with the equivalent loop of tapasin partially modeled projecting into the F pocket[18] (Fig. 1b, c-left), while X-ray structures of MHC-I/dipeptide complexes together with a short hydrophobic peptide occupying the groove, derived from the tapasin loop, appear to show a stable, bound conformation[19]. Recent studies have provided supporting evidence that the TAPBPR G24-R36 loop influences peptide exchange on the MHC-I, either by functioning as a peptide placeholder in the empty MHC-I/TAPBPR complexes[20] or by acting as a lever to promote dissociation of suboptimal peptides[21]. In contrast, mutational insults to the "scoop" loop have only minor effects on tapasin-mediated intracellular processing of nascent MHC-I[19]. In an independent crystal structure of the TAPBPR/MHC-I complex, density for the G24-R36 loop is lacking (Fig. 1c-right), presumably either due to inherent disorder of the loop or due to the presence of a disulfide-linked truncated peptide with large degrees of rotational freedom that could influence the loop's conformation[16]. Thus, while the TAPBPR G24-R36 loop seems to influence peptide exchange and antigen selection, there remain outstanding questions of whether it adopts a conformation that enters the empty MHC-I groove and how it affects peptide loading relative to the shorter tapasin loop.

Here, using a combination of in situ functional assays, yeast display/MHC-I tetramer staining, and a range of complementary biophysical techniques including solution NMR, we address the G24-R36 loop discrepancy between the two MHC-I/TAPBPR X-ray structures[16,17] and the MHC-I/Tapasin cryoEM structural model[18], and determine its role in the chaperoning and editing functions. Our findings suggest that the G24-R36 loop is not necessary for chaperoning activity, but directly participates in peptide editing on the MHC-I groove. The TAPBPR G24-R36 loop does not interact directly with the floor of the MHC-I groove to compete with incoming peptides, but instead hovers above the MHC-I groove to promote loading by acting as a trap for early peptide intermediates within the MHC-I/TAPBPR intermediate complex. Our study also suggests the possibility that there may be significant differences in how TAPBPR utilizes the G24-R36 loop to perform its peptide editing function across different MHC-I alleles due to unique steric or electrostatic chemical properties of the pMHC-I surface. Together, our results underscore a role for TAPBPR as an auxiliary editor that operates in compartments of reduced peptide concentration.

## Results

**Unexpected conformational plasticity of the TAPBPR G24-R36 loop**. A re-examination of the PLC cryo-EM structure[18] shows that the tapasin loop modeled as projecting into the F pocket falls outside the electron density and has B factors fixed at 30.00 Å$^2$, indicating it was excluded from refinement. Nearby density suggests that the loop instead rests above the peptide-binding groove (Fig. 1c). The TAPBPR G24-R36 loop was built in only one of the TAPBPR co-crystal structures where it was bound to the H-2D$^b$ murine MHC-I[17], and its assigned B factors were exceedingly high (Fig. 1c; PDB ID 5OPI), with additional large B factor transitions between bonded amino acids across the entire model. In the second crystal structure of TAPBPR-bound H-2D$^d$ [16], no clear density was observed for the 24–36 loop in the four different complexes of the asymmetric unit (Fig. 1c; PDB ID 5WER). The loop's conformation and placement relative to the empty MHC-I groove are therefore uncertain in the original structural data.

To investigate the structural plasticity of the TAPBPR G24-R36 loop in the MHC-I/TAPBPR complex, we used complementary computational approaches. We chose a common human MHC-I molecule frequently represented in Caucasian and other global populations, HLA-A*02:01, as our model system[22]. In the first approach, the HLA-A*02:01 sequence was threaded onto the H2-D$^d$/TAPBPR X-ray structure missing electron density for the G24-R36 loop (PDB ID 5WER)[16]. In our model, we replaced the cysteine mutation present in the initial H2-D$^d$/TAPBPR structure with the wild-type HLA-A*02:01 residue and did not include the truncated peptide in the groove, which is also lacking from the X-ray structure. The resulting peptide-deficient HLA-A*02:01/TAPBPR complex was used as a template to explore whether an α-helical conformation of the G24-R36 loop could be sampled during loop modeling. We employed a variety of sampling algorithms, including cyclic coordinate descent (CCD), kinematic closure (KIC), and comparative modeling (CM), within the Rosetta modeling suite[23]. The resulting lowest energy HLA-A*02:01/TAPBPR models revealed a wide range of TAPBPR G24-R36 loop conformations, each differing significantly from the α-helical conformation of the H2-D$^b$/TAPBPR X-ray structure (Supplementary Fig. 1a–c and Fig. 1a–c). In the second approach, the HLA-A*02:01 sequence was threaded onto the H2-D$^b$/TAPBPR X-ray structure containing the G24-R36 "scoop loop" modeled with poorly resolved electron density (PDB ID 5OPI). The modeled peptide-deficient HLA-A*02:01/TAPBPR complex was used as a starting point to determine whether the "scoop loop" conformation was stable in all-atom molecular dynamics (MD) simulations, performed in explicit solvent. To evaluate the stability of our MD simulations, we examined Cα root-mean-square deviation (RMSD) as a function of simulation time for the entire HLA-A*02:01/TAPBPR complex, as well as for the individual components of the system, relative to a global reference frame defined by the starting structure. The HLA-A*02:01/TAPBPR complex and its components exhibited Cα RMSD values that are in a similar range to those observed by other groups performing MD simulations on analogous systems[24,25] (Supplementary Fig. 1d). The Cα RMSD analysis verifies the global stability of the entire complex and reveals that our system has reached equilibrium to a stable energy basin.

The MD trajectory revealed a loss of helical propensity of the short α-helix formed by TAPBPR residues G24-L30, as well as the movement of the TAPBPR G24-R36 loop away from the HLA-A*02:01 groove within the first 50 ns. This was highlighted by a marked increase in Cα-Cα distances measured between HLA-A*02:01 groove (D77, T80, K146) and TAPBPR G24-R36 loop (A29, L30) residues over the course of the simulation (Fig. 1d, e). During the MD simulations, the TAPBPR G24-R36 loop sampled both an open state (pointing away from MHC-I groove) and a closed state that hovers above the MHC-I groove in an extended conformation (Fig. 1d, e). Notably, other loops of TAPBPR showed less conformational variation, suggesting conformational diversity is a unique property of the G24-R36 loop (Supplementary Fig. 1e). Our MD results suggest a lack of interactions between the TAPBPR G24-R35 loop and the MHC-I groove, albeit they are limited by the relatively short 200 ns timeframe, which is insufficient to sample loop movements on the medium to slow timescale[26]. Notwithstanding, our MD simulations and Rosetta modeling employ different energy functions, and yet the two approaches independently sampled similar, non-helical TAPBPR G24-R36 loop conformations (Supplementary Fig. 1a–c and Fig. 1d, e) which do not interact with the floor of the MHC-I groove.

**The G24-R36 loop of TAPBPR and its equivalent in tapasin are not essential for chaperoning MHC-I**. Tapasin was knocked out

in human Expi293F cells, causing diminished chaperone-dependent processing and trafficking of endogenous HLA-A*02:01 to the cell surface, as measured by flow cytometry (Supplementary Fig. 2a–h). Surface expression of HLA-A*02:01 was rescued by transfection with a plasmid encoding tapasin, forming the basis for a fluorescence-based assay and selection measuring the effects of tapasin mutations (Fig. 2a and Supplementary Fig. 2a–c). Based on structural alignments (Fig. 1b), residues 11-19 of the tapasin loop were replaced with 22-35 of TAPBPR, and the protein remained functional, albeit with decreased activity which may reflect a minor structural perturbation (Fig. 2a). G15 of tapasin resides at the very tip of the loop, and shortening the loop by two residues on either side (tapasin Δ2step) had no effect on tapasin activity (Figs. 1a and 2a). Shortening the loop further to remove L18 (tapasin Δ3step) caused a sudden drop in activity, although the protein still remained highly active. The cryo-EM density for the PLC[18] is consistent with tapasin L18 contacting the rim of the peptide-binding groove above Y84 of HLA-A*02:01. Targeted substitutions of tapasin L18, chosen to be highly disruptive based on altered side-chain properties, caused similar decreases in activity, whereas substitutions of G15 at the very tip of the loop had minimal effect.

Defective surface expression of endogenous HLA-A*02:01 in tapasin-knockout (tapasin-KO) cells was rescued by transfection of a tapasin library encompassing 2160 single substitutions at 108 sites. Cells with the highest surface HLA-A*02:01 were collected by FACS and deep sequenced to determine the enrichment or depletion of all tapasin variants in the library; this method is known as deep mutational scanning. As controls, mutations to 12 buried residues in the tapasin core were found to be generally depleted except for hydrophobic substitutions, whereas 8 surface residues distal from the MHC-I interface were mutationally permissive, with the exception of F218 which contacts ERp57 and is conserved[18]. The most highly conserved tapasin residues in the selection experiment were on the concave face of the N/IgV domain that rests below the MHC-I $\alpha_2$ domain, and at the hinge where the N/IgV and C/IgC domains meet. There is weaker sequence conservation dispersed across the C/IgC domain that interacts with the $\beta_2$m and $\alpha_3$ domains. Tapasin loop residues V10-K20 were highly tolerant of mutations except for V10 and L18, which strongly prefer hydrophobic amino acids and are appropriately positioned to interact with HLA-A*02:01 M138 and Y84 on the upper rims of the $\alpha_{2-1}$ and $\alpha_1$ helices, respectively. Overall, targeted and deep mutagenesis show that, for folding and processing of HLA-A*02:01, a long loop on the chaperone is not necessary, but binding contacts to the top of the MHC-I α-helices are important.

We refer to the functional replacement of tapasin with TAPBPR in tapasin-KO cells as a tapasin surrogate assay. Overexpression of TAPBPR partially rescued surface HLA-A*02:01, although a subset of cells retained HLA-A*02:01 intracellularly at greater levels (Fig. 2d and Supplementary Fig. 2b). Substitution of the TAPBPR transmembrane and cytosolic segments with those of tapasin (called TAPBPR-CT) increased TAPBPR-mediated rescue of surface HLA-A*02:01, despite substantially reduced protein expression (Fig. 2d). We hypothesize that TAPBPR-CT may be recruited to the TAP transporter via the tapasin C-terminal region (Supplementary Fig. 2a), though this remains undemonstrated. TAPBPR-CT was used as the background in which mutations to the luminal domains of TAPBPR were tested. Deletion of the TAPBPR loop tip (TAPBPR ΔALAS, removing the last turn of the helix modeled for the loop in PDB 5OPI) caused a small decrease in HLA-A*02:01 processing, and deletion of the entire TAPBPR 24–36 loop (ΔG24-R36) caused a further decrease in activity.

Using the tapasin surrogate assay as the basis for fluorescence-based selection, the TAPBPR 24–35 loop was deep mutationally scanned to assess the relative activities of all 240 single amino acid substitutions. Apart from the heavy depletion of nonsense mutations, the mutational landscape across the 24–35 loop is relatively featureless (Fig. 2e). Overall, these results demonstrate that the loop is not essential for chaperoning function, although some effects can be observed for different length and sequence variants.

To compare with other regions of TAPBPR that may be important, a larger library was constructed encompassing all single amino acid substitutions across 75 TAPBPR residues at the crystallographic interface with MHC-I[16], in addition to 10 surface residues on the opposite side and to 7 buried residues as controls. Following the selection of the TAPBPR library for high levels of surface HLA-A*02:01 expression, polar mutations to buried residues in the TAPBPR core were depleted, whereas surface residues distal from the MHC-I interface were mutationally tolerant (Fig. 2f). Similar to our observations for deep mutagenesis of tapasin, two regions of TAPBPR within the MHC-I interface were tightly conserved for activity: the concave face of the N/IgV domain that rests below the MHC-I $\alpha_2$ domain, and the edge of the C/IgC domain that bridges the $\beta_2$m and $\alpha_3$ domains (Fig. 2c, f). There was a preference for hydrophobic amino acids across the interface, with polar residues increasingly tolerated towards the interface periphery. Targeted alanine substitutions were tested to confirm that TAPBPR residues contacting the $\alpha_2$ underside (G212A and I261A) and $\alpha_3/\beta_2$m junction (S333A) were critical for mediating rescue of surface HLA-A*02:01 in tapasin-KO cells, whereas residues contacting the upper rim of the peptide groove (E102A and M122A) were not (Fig. 2g, h). Overall, TAPBPR chaperone activity is primarily encoded in sites that scaffold the correct folded architecture of the heavy chain with $\beta_2$m, consistent with our prior conclusions that TAPBPR chaperones nascent, improperly folded MHC-I substrates within the cell[5]. Notably, the G24-R36 loop is not critical for this activity.

**The loop affects TAPBPR recognition of folded pMHC-I in an allele-dependent manner.** Trafficking of HLA-A*02:01 to the plasma membrane will depend on chaperoning activity of TAPBPR, but not necessarily its editing function. To directly assess how mutations impact TAPBPR binding to folded pMHC-I molecules, as would occur during editing of the bound peptide, the extracellular domain of TAPBPR was displayed on the surface of yeast and binding to fluorescent pMHC-I tetramers was detected by flow cytometry (Supplementary Fig. 3a–l). Bound MHC-I was competed off by the addition of peptides in a dose-dependent manner that correlated with peptide affinities (Supplementary Fig. 3k, l). The TAPBPR 24–35 loop was mutated, and the yeast-displayed library was screened for tight binding to two MHC-I alleles: mouse H2-D^d and human HLA-A*02:01 (Supplementary Fig. 3i, j). No binding was observed for tetramers of a third allele, human HLA-A*01:01, consistent with TAPBPR having restricted recognition of folded pMHC-I allotypes[5]. At nearly all positions of the 24–35 loop there was enrichment of large, hydrophobic, and aromatic residues for increased binding to both H2-D^d and HLA-A*02:01 (Fig. 3a). The design of protein–protein interfaces has shown that large hydrophobic residues can promote binding through non-specific apolar interactions[27,28]. Instead, our attention was drawn to allele-specific differences that are unlikely to be due to generic mutational effects. For binding to H2-D^d, positions 28 and 30 near the center of the TAPBPR 24–35 loop have strong preferences for aliphatic or aromatic side chains, whereas all non-basic substitutions of R27 are enriched (excluding cysteine, which

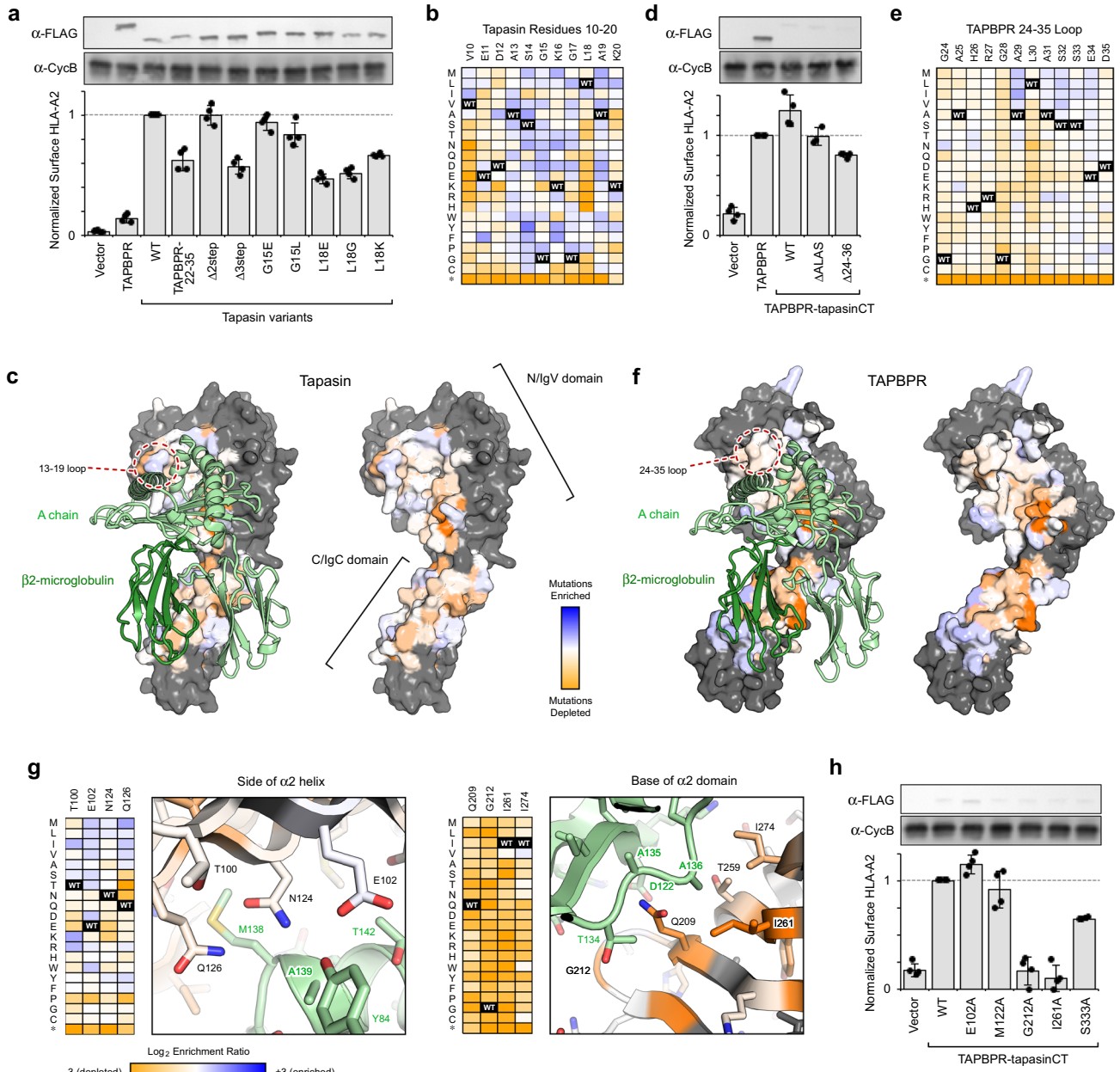

**Fig. 2 Important chaperone features for functional replacement of tapasin localize to scaffolding sites for MHC-I. a** TAPBPR and variants of tapasin were transfected in tapasin-KO Expi293F cells, and relative surface HLA-A*02:01 expression as measured by flow cytometry is plotted. Data are mean ± SD, $n = 4$ independent experiments. Proteins were FLAG-tagged at their luminal N-termini and immunoblots of the whole lysate are shown, with cyclophilin B (CycB) used as a loading control. **b** Tapasin was deep mutationally scanned based on selection of tapasin-KO cells with rescued surface HLA-A*02:01. Log$_2$ enrichment ratios for mutations across residues 10–20 are plotted from ≤−3 (depleted/deleterious, orange) to ≥+3 (enriched, dark blue). Residue position is on the horizontal axis, and amino acid substitutions are on the vertical axis (*, stop codon). **c** Conservation scores from the entire deep mutational scan of the tapasin/MHC-I interface are mapped to a homology model of HLA-A*02:01-bound tapasin. Highly conserved residues for mediating the folding and surface trafficking of HLA-A*02:01 are colored orange, while neutral regions are pale white/blue. Residues excluded from the library and analysis are gray. MHC-I H chain and hβ$_2$m are pale and dark green ribbons, respectively. **d** A chimera of the TAPBPR luminal domain with the tapasin TM and cytosolic domains, called TAPBPR-CT, has increased activity for chaperoning endogenous HLA-A*02:01, despite reduced protein expression by immunoblot (upper inset). Variants of TAPBPR were tested for rescue of surface HLA-A*02:01 in the TAPBPR-CT background. Data are mean ± SD, $n = 4$ independent experiments. **e** As in (**b**), now plotting log$_2$ enrichment ratios for mutations in TAPBPR residues 24–35 from a deep mutational scan for rescue of surface HLA-A*02:01 in TAPBPR library-transfected tapasin-KO cells. **f** As in (**c**), now showing sequence conservation from a deep mutational scan of TAPBPR mapped to a model of TAPBPR/HLA-A*02:01. **g** Close up views of two structural regions, colored as in (**f**). Accompanying heatmaps plot log$_2$ enrichment ratios for each mutation from depleted/deleterious (orange) to neutral (white and pale colors) to enriched (dark blue). **h** Individual mutations of TAPBPR-CT were validated by targeted mutagenesis in the tapasin surrogate assay. Data are mean ± SD, $n = 4$ independent experiments.

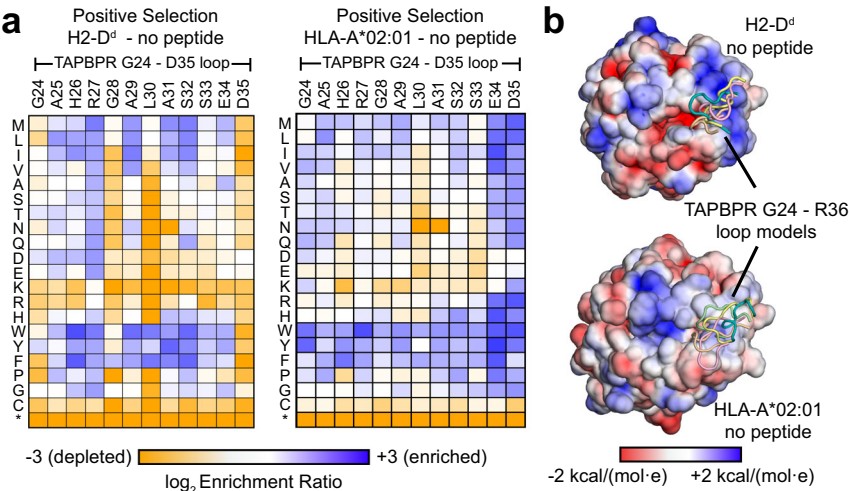

**Fig. 3 TAPBPR 24–35 loop sequence interaction landscape is pMHC-I allele-dependent. a** A TAPBPR library of substitution mutants in the 24–35 loop was displayed on the yeast surface and sorted for high binding signal to fluorescent pMHC-I tetramers made using refolded P18-I10/H2-D$^d$ and TAX9/HLA-A*02:01. In these experiments there was no excess of competing peptide; see Supplementary Fig. 3 for data from equivalent selections with competing free peptide. In the mutational landscapes, log$_2$ enrichment ratios for each mutation are plotted from ≤−3 (depleted/deleterious, orange) to ≥+3 (enriched, dark blue). TAPBPR 24–35 loop residue position is on the horizontal axis, and amino acid substitutions are on the vertical axis (*, stop codon). **b** Solvent-accessible surfaces of MHC-I molecules colored by electrostatic potential (negative in red, to positive in blue) as calculated in CHARMM-GUI PBEQ-Solver for H2-D$^d$ (PDB 3ECB) and HLA-A*02:01 (PDB 1DUZ). The five lowest energy RosettaCM models of the TAPBPR 24–35 loop are shown as references.

is deleterious at all positions). The data are therefore suggestive that the tip of the loop is near an apolar patch on H2-D$^d$ with a nearby electropositive charge; surfaces meeting these criteria can be found near the middle of the $\alpha_1$ helix (MHC-I residues V76 and R79) or at the end of the $\alpha_{2-1}$ helix (K146, A149, and A150). However, permissiveness in the landscape also suggests contacts between the 24–35 loop and H2-D$^d$ are sufficiently loose to tolerate mutations. Amino acid preferences within the 24–35 loop were weaker in the selections for binding HLA-A*02:01, although position 30 again prefers hydrophobic side chains. L30 has also been found to be important for binding to HLA-A*68:02[21]. TAPBPR D35 is tightly conserved as an acidic residue for high H2-D$^d$ binding due to electrostatic complementarity with basic residues (H2-D$^d$ R79 and R83) on the upper rim of the peptide-binding groove. HLA-A*02:01 has a glycine at position 79, and nearly all non-acidic substitutions of TAPBPR E34 and D35 enhance HLA-A*02:01 binding, consistent with Rosetta models of the 24–35 loop contacting the upper surfaces of the MHC-I $\alpha$-helices (Fig. 3b).

Selected mutations that increase or decrease MHC-I binding, predicted from the deep mutational scans, were validated by targeted mutagenesis (Supplementary Fig. 3i, j). Furthermore, the binding of yeast-displayed TAPBPR ΔALAS to tetramers of H2-D$^d$ or HLA-A*02:01 was only slightly reduced compared to wild-type (WT). TAPBPR ΔALAS and targeted point mutants exhibited competitive interactions with peptide (Supplementary Fig. 3k, l), indicating that differences in how tightly variants of the G24-R36 loop recognize MHC-I has no major bearing on peptide-mediated chaperone displacement. Rather, other sites of contact between TAPBPR and MHC-I must act as the sensors for high-affinity peptide binding within the groove. We further note that peptides of different sequences were slightly better or worse at displacing some TAPBPR mutants, indicating mutations in the G24-R36 loop have subtle, peptide sequence-specific effects.

**The TAPBPR G24-R36 loop does not interfere with the empty MHC-I groove**. TAPBPR mutants ΔALAS and ΔG24-R36 were recombinantly expressed in S2 *Drosophila* cells, purified, and

characterized in vitro. Circular dichroism (CD) spectroscopy showed that TAPBPR G24-R36 loop mutants retain the expected immunoglobulin (Ig)-like fold, as highlighted by the 2-sheet characteristic of a negative band between 215 and 219 nm (Fig. 4a, top). By differential scanning fluorimetry (DSF), TAPBPR G24-R36 loop mutants exhibited thermal stability comparable to WT with melting temperature ($T_m$) in the 48–49 °C range (Fig. 4a, bottom). Finally, both TAPBPR$^{WT}$ and TAPBPR$^{ΔG24-R36}$ proteins readily formed high-affinity complexes with empty H2-D$^d$ or HLA-A*02:01 molecules, prepared upon refolding with the photosensitive peptides photoP18-I10 and photoFluM1, respectively, upon UV-mediated peptide release (Supplementary Fig. 4a, b).

To investigate specific loop interactions with MHC-I molecules, we used solution NMR and characterized HLA-A*02:01/TAPBPR complexes prepared using either wild-type or mutant TAPBPR[5,29]. The 87 kDa molecular weight of the MHC-I/TAPBPR complex prompted the use of perdeuteration and isoleucine, leucine, and valine (ILV) methyl probes, whose resonances are more robust against the size of the system investigated due to their favorable relaxation properties[30]. We isolated 1:1 stoichiometric peptide-deficient HLA-A*02:01/TAPBPR complexes where the HLA-A*02:01 heavy chain was selectively ILV$^{proS}$ $^{13}$C/$^1$H methyl labeled on a $^{12}$C/$^2$H background, whereas h$\beta_2$m and TAPBPR were at natural isotopic abundance. These experiments allowed us to examine how the presence of the TAPBPR G24-R36 loop influences the chemical environment of the 37 ILV$^{proS}$ methyl probes on HLA-A*02:01, several of which are in the groove and within 10 Å from the TAPBPR G24-R36 loop in the proposed "scoop loop" conformation (Fig. 4b). A subset of the probes (L78 δ2, L81 δ2, and L156 δ2) were missing from the 2D $^1$H-$^{13}$C methyl HMQC spectra due to NMR line broadening resulting from conformational exchange in the intermediate (μs-ms) timescale (Fig. 4b, gray spheres). A comparison of chemical shifts of the remaining probes did not identify any noticeable changes in 2D $^1$H-$^{13}$C methyl HMQC spectra of peptide-deficient HLA-A*02:01/TAPBPR complexes prepared with TAPBPR$^{WT}$, TAPBPR$^{ΔALAS}$, or TAPBPR$^{ΔG24-R36}$ (Fig. 4c). These results were corroborated by similar observations

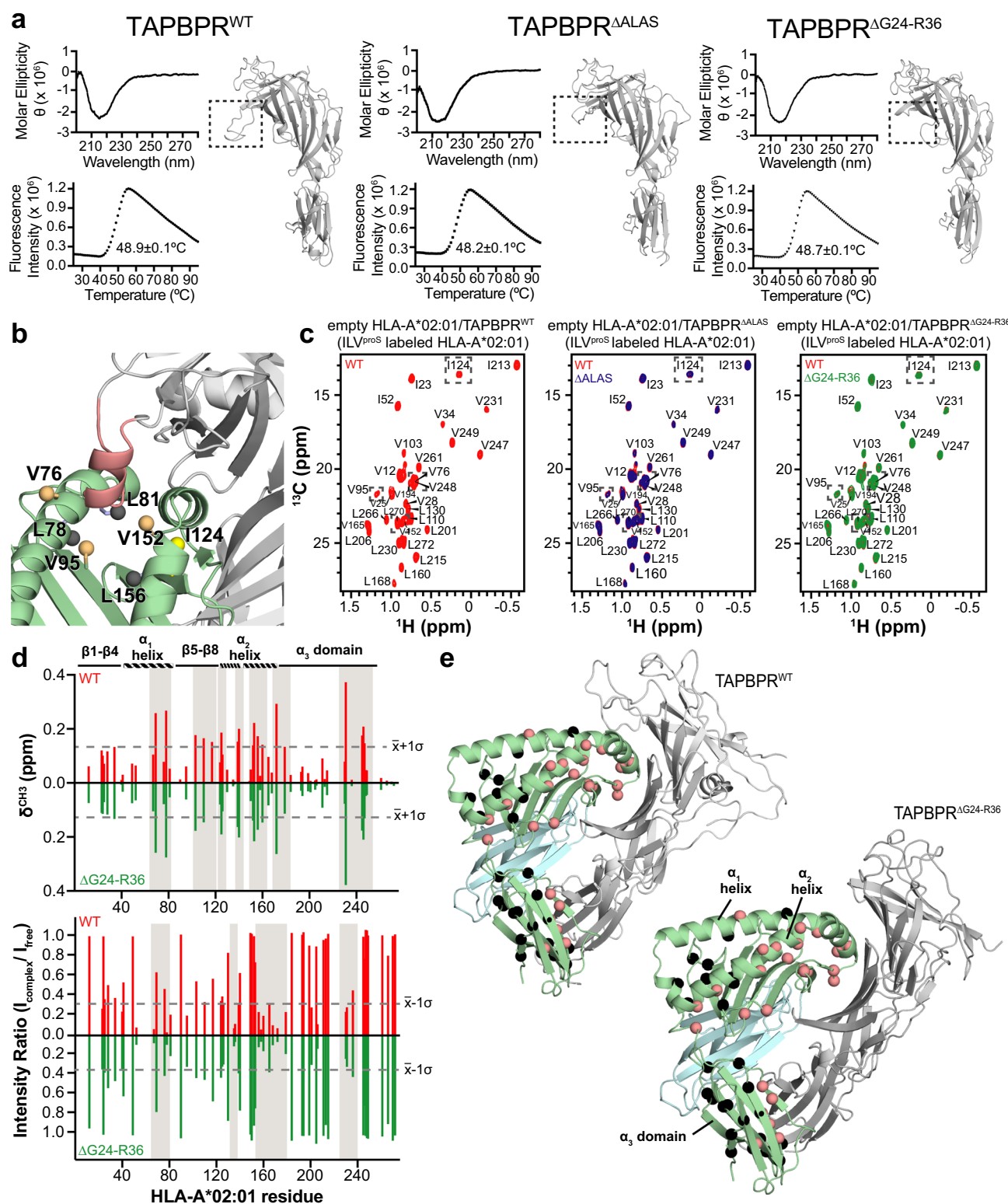

using independently prepared peptide-deficient MHC-I/TAPBPR complexes employing a more general methyl labeling scheme (AILV) on HLA-A*02:01, where the number of total methyl probes increased to 94, 20 of which were within 10 Å from the TAPBPR G24-R36 loop (Supplementary Fig. 5a). Chemical shift deviation (CSD) and intensity ratio analysis of AILV methyl resonances of empty HLA-A*02:01 bound to either TAPBPR^WT or TAPBPR^ΔG24-R36 relative to the resonances of unbound TAX9/HLA-A*02:01, revealed that HLA-A*02:01 methyl probes

affected upon complex formation with TAPBPR are not influenced by the removal of the TAPBPR G24-R36 loop (Fig. 4d). Mapping our NMR results on RosettaCM models of the HLA-A*02:01/TAPBPR complex revealed a similar interaction surface as observed previously for other MHC-I/chaperone complexes by solution NMR, X-ray crystallography, and cryoEM[16–18,29], suggesting that TAPBPR forms a stable complex with MHC-I molecules independently of the TAPBPR G24-R36 loop (Fig. 4e). We also examined chemical shift changes in

**Fig. 4 The TAPBPR G24-R36 loop does not enter the HLA-A*02:01 groove. a** Comparison between wild-type (WT) and mutant TAPBPR constructs used in this study. ΔALAS = deletion of residues A29-S32 and ΔG24-R36 = deletion of residues G24-R36. (Top) Far-UV CD and (Bottom) DSF spectra of each TAPBPR construct. Data are mean for $n = 3$ technical replicates. An inset in the DSF spectra notes measured thermal melt ($T_m$) values. The corresponding RosettaCM model of each TAPBPR construct is shown. The dotted box highlights differences in the G24-R36 loop region. **b** View of the peptide-deficient HLA-A*02:01/TAPBPR model (template PDB ID 5OPI) showing ILV$^{proS}$ methyl probes on HLA-A*02:01 (as spheres) within 10 Å of the TAPBPR G24-R36 loop (salmon). Methyl resonances of residues L78, L81 and L156 (shown in black) are missing in 2D $^1$H-$^{13}$C methyl HMQC spectra of peptide-deficient HLA-A*02:01/TAPBPR complex due to conformational exchange induced line broadening. **c** 2D $^1$H-$^{13}$C methyl HMQC spectra of 80 μM peptide-deficient HLA-A*02:01 (ILV$^{proS}$ labeled)/h$\beta_2$m in complex with TAPBPR$^{WT}$ (red), TAPBPR$^{\Delta ALAS}$ (blue) or TAPBPR$^{\Delta G24-R36}$ (green) recorded at 25 °C at a $^1$H field of 800 MHz. Dotted boxes highlight methyl probes (shown in panel (**b**)) that are modeled to be near the TAPBPR G24-R36 loop. As a reference, the 2D $^1$H-$^{13}$C methyl HMQC spectra of the complex formed with TAPBPR$^{WT}$ (red) is superimposed on those formed with TAPBPR$^{\Delta ALAS}$ or TAPBPR$^{\Delta G24-R36}$. **d** Comparison of chemical shift deviations (CSD, $\delta^{CH3}$, ppm) (top) and intensity changes ($I_{complex}/I_{free}$) (bottom) for AILV methyl probes of peptide-deficient HLA-A*02:01 in complex with either TAPBPR$^{WT}$ (red) or TAPBPR$^{\Delta G24-R36}$ (green), relative to unbound TAX9/HLA-A*02:01. Dotted lines represent the average plus one standard deviation for CSD analysis ($\bar{x} + 1\sigma$) or minus one standard deviation for intensity ratio analysis ($\bar{x} - 1\sigma$). Gray boxes highlighted affected regions of HLA-A*02:01. The protein domains of HLA-A*02:01 are shown for reference. **e** Mapping of affected HLA-A*02:01 methyl probes exhibiting either CSD or intensity changes upon complex formation with TAPBPR$^{WT}$ or TAPBPR$^{\Delta G24-R36}$ shown on RosettaCM models of the respective complex. Affected residues are colored salmon, unaffected residues are colored black.

HLA-A*02:01/TAPBPR complexes transiently bound to unlabeled TAX9 peptide in the presence of excess TAPBPR, to minimize TAPBPR dissociation from the pMHC-I. This allowed us to perform a full analysis of methyl groups along the MHC-I groove, since resonances previously broadened in the spectra of peptide-deficient complexes can now be directly observed (Supplementary Fig. 5b). Chemical shift analysis of the AILV methyl probes in wild-type and mutant TAX9/HLA-A*02:01/TAPBPR complexes did not reveal any measurable changes in 2D $^1$H-$^{13}$C HMQC spectra (Supplementary Fig. 5c). Together, these findings support the lack of a direct, stable interaction between the TAPBPR G24-R36 loop and the floor of the HLA-A*02:01 groove either in the empty or peptide-bound intermediate states.

**The TAPBPR loop promotes peptide loading on empty chaperoned MHC-I.** Our group has previously characterized the full thermodynamic cycle of TAPBPR-mediated MHC-I peptide exchange[29]. The cycle is defined by dissociation constants ($K_D$) for four reversible steps: association of TAPBPR with MHC-I in the absence of peptide ($K_{D1}$), association of peptide with MHC-I in the absence of TAPBPR ($K_{D2}$), association of pMHC-I with TAPBPR ($K_{D3}$), and association of peptide with the MHC-I/TAPBPR complex ($K_{D4}$) (Fig. 5a). We measured apparent $K_D$ values focusing on specific steps of the cycle using suitable isothermal titration calorimetry (ITC) experiments. ITC was performed on a range of known HLA-A*02:01 peptides (TAX8 - LFGYPVYV, TAX9 - LLFGYPVYV and KLL15 - KLLEIPDPDKNWATL) to uncover any differences in the TAPBPR G24-R36 loop effect as a function of peptide length[31,32]. Optimization of ITC experimental conditions (see "Methods") allowed us to focus on different steps of the thermodynamic cycle (Fig. 5b and Supplementary Fig. 6a–c). First, direct measurement of the binding of peptides to the empty MHC-I groove ($K_{D2}$) is hindered by the inherent instability of unchaperoned peptide-deficient MHC-I molecules. Thus, we utilized TAPBPR as a stabilizer of peptide-deficient HLA-A*02:01. Purified 1:1 HLA-A*02:01/TAPBPR complex in the calorimeter cell was titrated by injecting peptide. The $K_{D2}$ ($K_{D2,app}$) can be obtained from the experiment when peptide binds to the MHC-I and promotes the release of TAPBPR under conditions where no excess TAPBPR is included (Fig. 5b-left and Supplementary Fig. 6a). The $K_{D2,app}$ values, which range from 40 to 400 nM across our peptide set, did not differ significantly between HLA-A*02:01/TAPBPR complexes prepared using TAPBPR$^{WT}$ or TAPBPR$^{\Delta G24-R36}$, suggesting that our measurements were dominated by peptide binding to empty, unchaperoned MHC-I (Fig. 5c and Table 1). Our determined $K_{D2}$ values are consistent with previous reports of low

to medium nM affinity binding for peptide to MHC-I, confirming that our assay provides a close approximation for $K_{D2}$[33,34].

By titration of TAPBPR with pMHC-I in the presence of excess peptide in both the cell and syringe, the $K_{D3}$ for the formation of the ternary pMHC-I/TAPBPR complex was obtained (Fig. 5b-middle and Supplementary Fig. 6a). Measured $K_{D3}$ values, which were in the range of 2 μM for the different peptides in our set, were similar between pMHC-I/TAPBPR complexes formed using TAPBPR$^{WT}$ or TAPBPR$^{\Delta G24-R36}$ (Fig. 5c and Supplementary Fig. 6a–c and Table 1), suggesting that the loop does not affect recognition of peptide-loaded MHC-I. Finally, titration of empty HLA-A*02:01/TAPBPR complex by injecting the peptide in the presence of excess TAPBPR in both the cell and syringe (to minimize dissociation of TAPBPR from the pMHC-I/TAPBPR complex) allowed measurement of an apparent value for $K_{D4}$, binding of the peptide to an empty MHC-I/TAPBPR complex (Fig. 5b-right and Supplementary Fig. 6b). Here, we observed binding to two sequential sites (Fig. 5b-right), due to a combination of processes that are described by the steps indicated with $K_{D2}$ and $K_{D4}$ (Fig. 5a). However, given that the $K_{D2}$ process does not involve TAPBPR, results obtained in these conditions can be used to probe effects of TAPBPR loop variations on $K_{D4}$.

Notably, $K_{D4}$ values were increased by a factor of two to threefold between pMHC-I/TAPBPR complexes prepared using TAPBPR$^{\Delta G24-R36}$ versus TAPBPR$^{WT}$ (Fig. 5a–c and Supplementary Fig. 6a–c and Table 1); i.e., the G24-R36 loop increases the affinity of empty, chaperoned MHC-I for incoming peptides. While the instability of peptide-deficient HLA-A*02:01 molecules in the absence of TAPBPR did not permit direct measurement of $K_{D1}$ (i.e., binding of TAPBPR to empty MHC-I), this value was inferred from thermodynamic balance along the exchange cycle (Fig. 5a and Table 1). $K_{D1}$ was decreased by a factor of two to threefold for TAPBPR$^{\Delta G24-R36}$ versus TAPBPR$^{WT}$ (Table 1), suggesting that the presence of the loop reduces the affinity of TAPBPR for empty MHC-I, likely due to steric clashes between the TAPBPR loop and the MHC-I groove (Supplementary Fig. 1a–c and Fig. 1d, e). This result is inconsistent with models of the TAPBPR 24–36 loop acting as a stabilizer of empty MHC-I molecules and a steric competitor of incoming peptides[20,21]. In agreement with our NMR results (Fig. 4c and Supplementary Fig. 5a, c), the ITC data suggest that the TAPBPR G24-R36 loop does not form a strong, stabilizing interaction with the empty MHC-I groove but instead functions to promote peptide loading to the empty MHC-I/TAPBPR complex.

**The TAPBPR loop length is important for peptide loading function in vitro.** To determine whether the full length of the

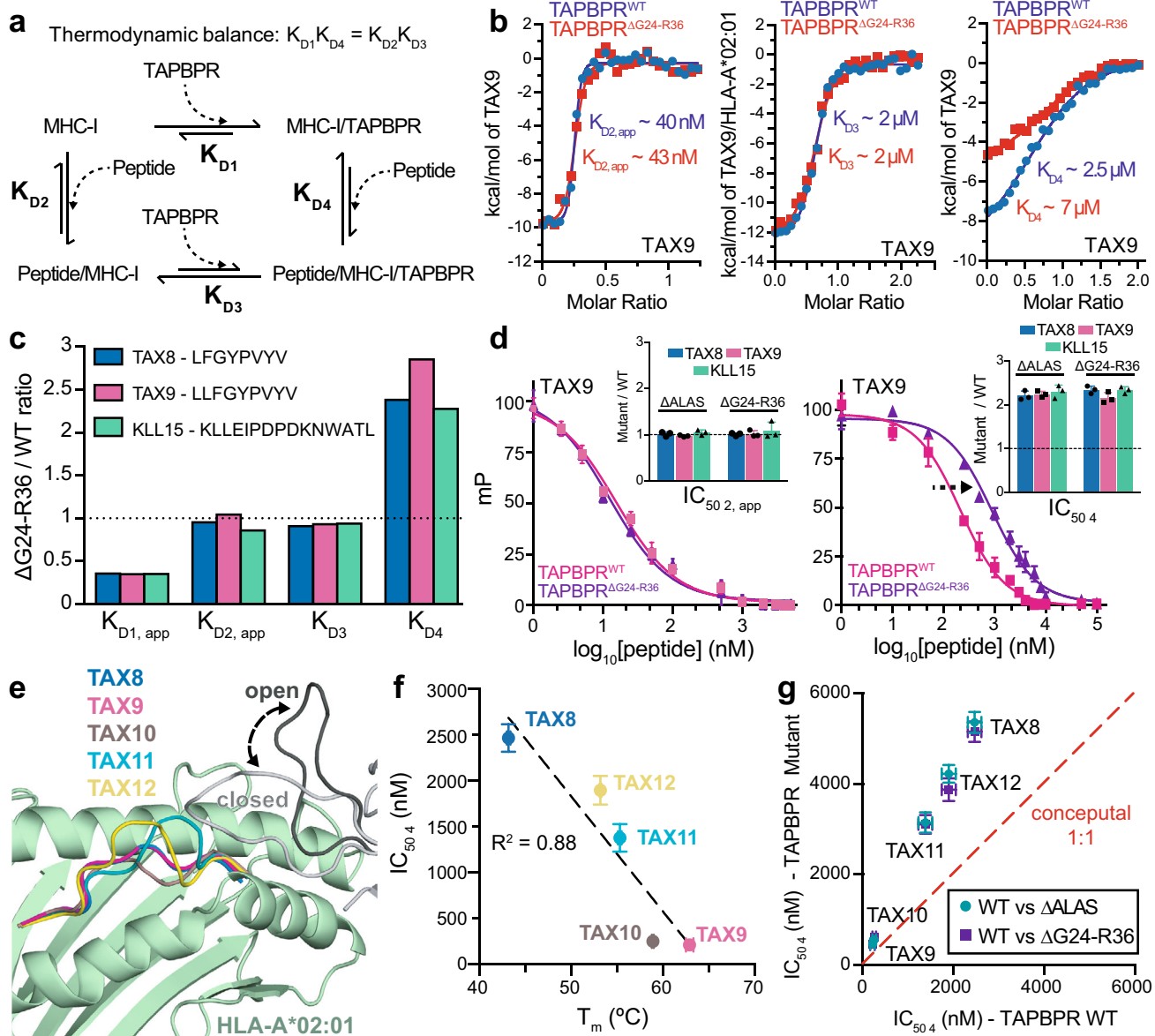

**Fig. 5 The TAPBPR G24-R36 loop promotes peptide binding on empty MHC-I. a** Schematic of the TAPBPR-mediated MHC-I peptide exchange cycle. The dissociation constant ($K_D$) of each step is noted. **b** ITC performed at different stages of the peptide exchange cycle ($K_{D2}$, $K_{D3}$, $K_{D4}$) for TAX9/HLA-A*02:01/hβ2m with TAPBPR[WT] or TAPBPR[ΔG24-R36] (see "Methods"). $K_{D1}$ was not measured directly, but inferred from thermodynamic balance along the cycle shown in (**a**). Due to experimental limitations, the measured $K_{D1}$ and $K_{D2}$ are "apparent" and thus denoted with $K_{D1,app}$ and $K_{D2,app}$. $K_D$ and n values determined by ITC are noted in Table 1. **c** Apparent $K_D$ ratios determined by ITC for TAPBPR[ΔG24-R36]/TAPBPR[WT] for TAX8, TAX9, and KLL15 peptides. The dotted line represents no effect. **d** FP performed under substoichiometric (left) and stoichiometric (right) conditions. Millipolarization (mP) values are plotted as a function of the $\log_{10}$ peptide concentration for TAX8, TAX9, and KLL15 competitor peptides. The insets show a comparison of the ratio of FP-determined $IC_{50}$ values for TAPBPR[ΔG24-R36] or TAPBPR[ΔALAS] versus TAPBPR[WT]. The dotted line represents no effect. **e** X-ray structures (TAX8, TAX9) and RosettaCM models (TAX10, TAX11, TAX12) show peptide bulging within the HLA-A*02:01 groove. The open and closed conformation of the TAPBPR G24-R36 loop from MD simulations are shown in gray and black, respectively. **f** Comparison of FP-determined $IC_{50}$ values under stoichiometric conditions for TAX length variants versus pMHC-I thermal stability. The $R^2$ value of the linear regression fit (black line) is shown. **g** Comparison of FP-determined $IC_{50}$ values under stoichiometric conditions for TAX length variants for TAPBPR[WT] versus TAPBPR[ΔALAS] or TAPBPR[ΔG24-R36]. The dotted red line represents a conceptual 1:1 correlation (no effect). Data presented in panels (**d**), (**f**), and (**g**) are mean ± SD for $n = 3$ independent experimental replicates.

TAPBPR G24-R36 loop is required for normal function, we applied a similar design approach from our ITC studies to fluorescence polarization (FP) experiments. We have previously shown that the association of peptides with the MHC-I groove can be monitored with FP competition assays where purified peptide-deficient MHC-I/TAPBPR complexes are incubated with fluorescently labeled TAMRA-peptide and a range of concentrations of unlabeled competitor peptide[29]. Analogous to our ITC experiments, optimization

of FP assay conditions (see "Methods") allowed us to independently probe each step of the thermodynamic cycle (Supplementary Fig. 7a–c). First, we explored the dilution of peptide-deficient HLA-A*02:01/TAPBPR complexes to substoichiometric conditions. Here, the conditions are considered substoichiometric because the concentration of HLA-A*02:01/TAPBPR complex used (~50 nM) is below the estimated $K_D$ of the MHC-I/TAPBPR interaction (~190 nM), previously determined by SPR experiments of an

**Table 1 ITC derived apparent dissociation constants for each step of the TAPBPR-mediated peptide exchange cycle.**

|  | TAPBPR$^{WT}$ | *n* value | TAPBPR$^{\Delta G24-R36}$ | *n* value |
|---|---|---|---|---|
| TAX8 $K_{D1,app}$ | 37.4 nM | n/d | 13.9 nM | n/d |
| TAX9 $K_{D1,app}$ | 30.6 nM | n/d | 11.1 nM | n/d |
| KLL15 $K_{D1,app}$ | 38.8 nM | n/d | 14.2 nM | n/d |
| TAX8 $K_{D2,app}$ | 326 ± 14 nM | 0.43 ± 0.02 | 316 ± 19 nM | 0.44 ± 0.03 |
| TAX9 $K_{D2,app}$ | 40.2 ± 6.5 nM | 0.42 ± 0.01 | 42.6 ± 8.3 nM | 0.46 ± 0.01 |
| KLL15 $K_{D2,app}$ | 383 ± 22 nM | 0.45 ± 0.06 | 334 ± 20 nM | 0.41 ± 0.01 |
| TAX8 $K_{D3}$ | 2.6 ± 0.5 µM | 1.00 ± 0.03 | 2.4 ± 0.6 µM | 1.03 ± 0.01 |
| TAX9 $K_{D3}$ | 1.9 ± 0.7 µM | 1.02 ± 0.01 | 1.8 ± 0.3 µM | 1.06 ± 0.02 |
| KLL15 $K_{D3}$ | 2.2 ± 0.5 µM | 1.05 ± 0.04 | 2.1 ± 0.3 µM | 1.03 ± 0.01 |
| TAX8 $K_{D4}$ | 22.7 ± 8.2 µM | 1.01 ± 0.02 | 54.4 ± 9.4 µM | 1.05 ± 0.02 |
| TAX9 $K_{D4}$ | 2.5 ± 0.8 µM | 1.04 ± 0.01 | 7.3 ± 0.7 µM | 1.02 ± 0.02 |
| KLL15 $K_{D4}$ | 21.6 ± 7.0 µM | 1.10 ± 0.05 | 49.5 ± 8.6 µM | 1.03 ± 0.04 |

TAX8 = LFGYPVYV
TAX9 = LLFGYPVYV
KLL15 = KLLEIPDPDKNWATL
WT = wild-type
Due to experimental limitations, the measured $K_{D1}$ and $K_{D2}$ values are "apparent", denoted with the use of "app".
n/d not determined because the $K_{D1}$ value is calculated from thermodynamic balance, *n* value corresponds to observed stoichiometry by ITC.

analogous system[16]. Thus, in the absence of excess TAPBPR, peptide promotes the dissociation of HLA-A*02:01 from TAPBPR, which allows for measurement of peptide apparent IC$_{50\,2}$ (IC$_{50\,2,app}$) values. FP was performed with TAX8, TAX9, and KLL15 serving as the competitor peptides. In agreement with ITC, IC$_{50\,2,app}$ values determined by FP did not differ between HLA-A*02:01/TAPBPR complexes prepared using TAPBPR$^{WT}$, TAPBPR$^{\Delta ALAS}$, or TAPBPR$^{\Delta G24-R36}$ (Fig. 5d, left and Supplementary Fig. 7b). Second, titration of TAPBPR into a sample containing fluorescently labeled TAMRA-TAX9 in complex with HLA-A*02:01 allows measurement of K$_{D3}$ in the peptide exchange cycle (Fig. 5a). In agreement with our ITC experiments, we find that measured K$_{D3}$ values from FP, which are in a similar range to ITC of 2 µM, did not differ between pMHC-I/TAPBPR complexes formed using TAPBPR$^{WT}$, TAPBPR$^{\Delta ALAS}$, or TAPBPR$^{\Delta G24-R36}$ (Supplementary Fig. 7a). Finally, dilution of peptide-deficient HLA-A*02:01/TAPBPR complexes under stoichiometric conditions (in the presence of excess TAPBPR to minimize dissociation of TAPBPR from the pMHC-I/TAPBPR complex) allows for measurement of IC$_{50\,4}$. In agreement with our ITC data, we find that measured IC$_{50\,4}$ values are increased by a factor of two to threefold between pMHC-I/TAPBPR complexes formed using ΔG24-R36 versus wild-type TAPBPR (Fig. 5d, right and Supplementary Fig. 7c). In summary, the combined ITC and FP results suggest that the TAPBPR G24-R36 loop influences the formation of the pMHC-I/TAPBPR intermediate complex. These results further suggest that, because the short TAPBPR$^{\Delta ALAS}$ deletion variant already has the same behavior as TAPBPR$^{\Delta G24-R36}$, a full-length TAPBPR G24-R36 loop is an important factor for peptide exchange.

**The TAPBPR loop does not explicitly control the length of bound peptides.** We applied our FP experiments to determine whether the length of the MHC-I cargo influences TAPBPR G24-R36 loop-mediated peptide selection. Since it is established that peptides of length longer than ten bulge out of the MHC-I groove[32,35], we hypothesized that steric clashes between protruding peptide residues with the G24-R36 loop, which hovers above the groove, could influence TAPBPR-mediated peptide exchange. To test our hypothesis, we prepared a set of peptides derived from the HTLV-1 TAX epitope ranging from 8 to 12 amino acids in length, each containing the same residue types in the two anchor positions. Comparison of X-ray structures available for TAX8 and TAX9 with RosettaCM models of TAX10 (LLFGGYPVYV), TAX11 (LLFGGGYPVYV), and TAX12 (LLFGGGGYPVYV) revealed the expected trend with longer

peptides bulging from the HLA-A*02:01 groove (Fig. 5e). Using DSF experiments we observed that optimal peptide length (9–10 residues[36]) correlates with the highest thermal stability of the pMHC-I complex, as shown by the low stability of TAX8 and TAX12 ($T_m = 43.2$ and $53.2\,°C$) and higher stability for TAX9 and TAX10 ($T_m = 62.9$ and $58.9\,°C$) (Supplementary Fig. 8a). Next, we performed FP competition experiments under sub-stoichiometric conditions for each of the TAX length variants, to quantify peptide binding to empty MHC-I molecules upon release from TAPBPR. We observed a similar trend for FP-determined IC$_{50\,2,app}$ values, where TAX9 and TAX10 peptides exhibited stronger association with HLA-A*02:01 (Supplementary Fig. 8b). In addition, as observed in our ITC and previous FP experiments, measured IC$_{50\,2}$ values were similar between HLA-A*02:01/TAPBPR complexes prepared using TAPBPR$^{WT}$, TAPBPR$^{\Delta ALAS}$, or TAPBPR$^{\Delta G24-R36}$ (Supplementary Fig. 8b). We observed an excellent correlation between FP-determined IC$_{50\,4}$ values and $T_m$ values of pMHC-I complexes (Fig. 5f). In addition, we measured IC$_{50\,4}$ values for the TAX peptide length variants and found an increase by a factor of 2–3 between pMHC-I/TAPBPR complexes prepared using TAPBPR$^{\Delta ALAS}$ or TAPBPR$^{\Delta G24-R36}$ versus TAPBPR$^{WT}$, consistently across all peptides (Fig. 5g and Supplementary Fig. 8c). It is worth noting that the 15mer KLL15 peptide exhibits a similar magnitude in the G24-R36 loop effect compared to our TAX length variant dataset (Fig. 5d, g). Together, these data suggest that the TAPBPR G24-R36 loop promotes loading of moderate and high-affinity peptides, irrespective of their length.

**The TAPBPR loop promotes the formation of a transiently bound peptide state.** To gain a high-resolution view of how the TAPBPR G24-R36 loop might affect the binding of peptides on the MHC-I groove in an aqueous environment, we turned to NMR experiments using $^{13}$C-LV (Leu/Val) methyl-labeled peptides. We first examined a high-affinity TAX9 peptide, containing two N-terminal Leu and two C-terminal Val methyl groups which act as sensitive probes of the dynamics and local magnetic environment using samples where the MHC and TAPBPR components are at natural isotopic abundance (Fig. 6a). In the 2D $^1$H-$^{13}$C methyl HMQC spectra of pMHC-I refolded with labeled peptide and followed by addition of 8-fold molar excess TAPBPR, we observed NMR resonances corresponding to the free peptide, unloaded from the MHC-I groove by TAPBPR (denoted "f"), in slow exchange (ms timescale) with the MHC-I bound form

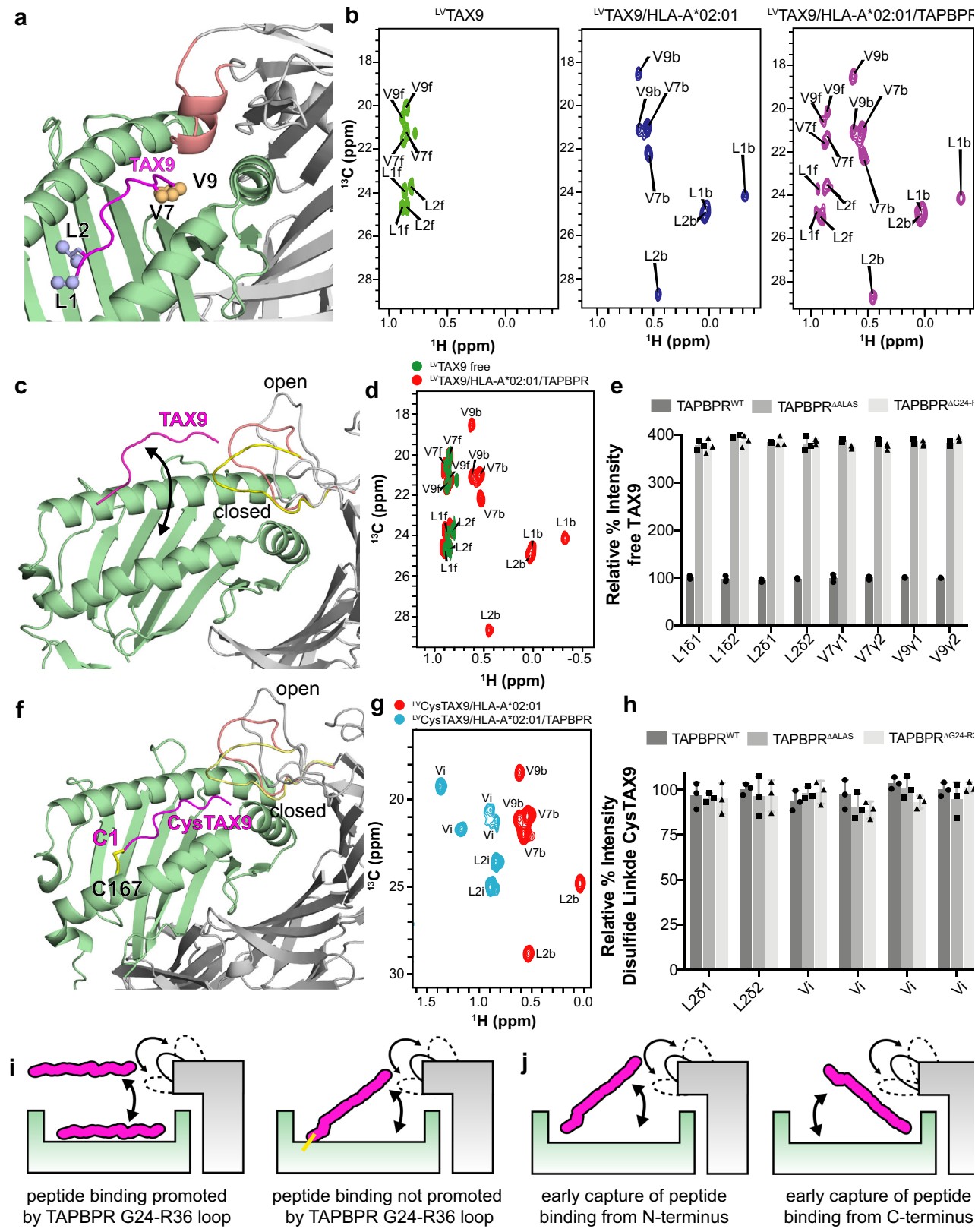

(denoted "b") (Fig. 6b). Resonances corresponding to bound peptide in the MHC-I/TAPBPR complex are broadened beyond detection, likely due to conformational exchange at the intermediate (μs-ms) timescale. To examine the effects on the TAPBPR G24-R36 loop on exchange, we developed an in vitro

NMR-based peptide unloading assay. The NMR signal intensities of methyl resonances corresponding to free TAX9 provided us a robust quantification of the amount of peptide in solution when pMHC-I is incubated with saturating concentration (eightfold molar excess) of TAPBPR, expressed with and without the

**Fig. 6 NMR characterization of chaperone-mediated peptide exchange on MHC-I using selective methyl $^{13}$C-labeled peptides. a** Rosetta model of the TAX9/HLA-A*02:01/TAPBPR complex using template PDB ID 5OPI. The Leu/Val methyl groups of TAX9 are shown as spheres. HLA-A*02:01 is green, TAX9 is magenta, TAPBPR is gray, and the TAPBPR "scoop loop" is colored salmon. **b** 2D $^{1}$H-$^{13}$C methyl HMQC spectra of 100 μM $^{15}$N/$^{13}$C LV-labeled TAX9 peptide in the free state (left, green), in complex with HLA-A*02:01/hβ2m (middle, blue), and in complex with HLA-A*02:01/hβ2m in the presence of saturating concentration (eightfold molar excess) TAPBPR (right, purple). All spectra were recorded at 25 °C at a $^{1}$H field of 800 MHz. Assignments of methyl resonances corresponding to the free and MHC-bound peptide states are denoted with 'f' and 'b', respectively **c** Rosetta model of the TAX9/HLA-A*02:01/TAPBPR complex using template PDB ID 5WER is shown highlighting that TAPBPR promotes the release of TAX9 peptide. The open, intermediate, and closed conformations of the TAPBPR G24-R36 loop obtained from MD simulations are shown in gray, salmon, and yellow. **d** Overlay of 2D $^{1}$H-$^{13}$C methyl HMQC spectra from 'A' of: $^{13}$C-LV-labeled TAX9 in the free state (green), and in complex with HLA-A*02:01/hβ2m in the presence of saturating concentration of TAPBPR (red). **e** Comparison of signal intensities for methyl resonances of TAX9 when released from the MHC-I/TAPBPR complex formed with and without the TAPBPR G24-R36 loop. **f** Rosetta model of disulfide-linked CysTAX9/HLA-A*02:01 in complex with TAPBPR designed using Disulfide by Design v2. The disulfide is formed between C1 of CysTAX9 (CLFGYPVYV) and W167C of HLA-A*02:01. The open and closed conformations of the TAPBPR G24-R36 loop obtained from MD simulations are shown in gray, salmon and yellow. **g** 2D $^{1}$H-$^{13}$C methyl HMQC spectra of $^{13}$C-LV-labeled CysTAX9 in complex with HLA-A*02:01/hβ2m (red) and HLA-A*02:01/ hβ2m in the presence of 8-fold molar excess TAPBPR (blue) recorded at 25 °C at a $^{1}$H field of 600 MHz. Methyl resonances of the pMHC-I bound state are denoted "b" for bound. Methyl resonances of the pMHC-I/ TAPBPR state are denoted "i" for intermediate, corresponding to a free peptide conformation that is still tethered to the MHC groove. **h** Comparison of NMR signal intensities for each methyl resonance of CysTAX9 in the pMHC-I/TAPBPR intermediate state, formed with and without the TAPBPR G24-R36 loop. The intermediate states for the Valines were not assigned. **i** NMR-based model of how the TAPBPR G24-R36 loop influences peptide loading on pMHC-I. **j** Two putative mechanisms for how the TAPBPR G24-R36 loop promotes the early capture of the peptide within the MHC-I groove with intermediate annealing from the N-terminus (left) or C-terminus (right). Data presented in panels (**e**) and (**h**) are mean ± SD for $n = 3$ independent experimental replicates.

G24-R36 loop (Fig. 6c, d). We found that incubation of pMHC-I with TAPBPR containing G24-R36 loop deletions resulted in ~3 to 4-fold increase of free TAX9 peptide relative to samples containing the equivalent concentrations of wild-type TAPBPR (Fig. 6e), suggesting that a longer loop results in reduced peptide unloading activity, consistent with our previous FP and ITC results (Fig. 5).

We hypothesized that the longer TAPBPR G24-R36 loop might act by promoting the capture of a transiently formed encounter complex between the peptide and the empty, chaperoned MHC-I groove (Fig. 6c). To test this hypothesis and to identify a plausible peptide loading pathway, we introduced a disulfide bond linking the N-terminus of isotopically $^{13}$C LV-labeled CysTAX9 peptide (L1C mutation) to the MHC-I groove (W167C mutation), which serves to mimic a transiently bound peptide state (Fig. 6f). This allowed us to observe a new set of methyl resonances, corresponding to an intermediate state with an N-terminally tethered peptide which lies outside of the groove (denoted "i"), in slow exchange with the resonances corresponding to the fully bound peptide (Fig. 6g). Contrary to our results using free, isotopically labeled peptides (Fig. 6e), a comparison of the intensities of methyl resonances reveals that tethering the peptide to the MHC-I groove alleviates the effect of the TAPBPR G24-R36 loop deletion in our NMR-based assay, resulting in a similar amount of peptide outside of the groove relative to WT TAPBPR (Fig. 6h, i). Since incoming peptides may anneal to the empty MHC groove via either their N-terminal or C-terminal anchors, TAPBPR may promote loading by stabilizing different ensembles of intermediate states with partially bound peptide conformations (Fig. 6j). If the dominant pathway for loading proceeded from the peptide N- to C- terminus then mutations of the TAPBPR G24-R36 loop would lead to a significant signal increase for methyl resonances corresponding to the intermediate state "i", contrary to our results (Fig. 6h). Therefore, TAPBPR likely acts by stabilizing an ensemble of peptide conformations forming transient interactions with the F-pocket of the MHC-I groove, adjacent to the TAPBPR loop. Similar results were obtained using an isotopically labeled 15mer peptide (Supplementary Fig. 9a–f and Supplementary Fig. 10a–h), demonstrating that the TAPBPR G24-R36 loop employs a similar mechanism to promote peptide capture irrespective of peptide length.

## Discussion

Recent structural studies have revealed that chaperone-mediated peptide exchange/editing is achieved through (i) stabilization of the peptide-deficient MHC-I in a peptide-receptive conformation, (ii) ejection of suboptimal peptides by inducing an ~3 Å widening of the MHC-I groove, and (iii) regulation of a dynamic switch located in the MHC-I groove[5,16,17,29]. Despite these important findings, the mechanistic details of how chaperones influence the repertoire of MHC-I presented antigens remains incompletely characterized, while a significant difference between published X-ray structures challenges the proposed role of the TAPBPR G24-R36 loop as a direct peptide competitor.

Here, we resolve the key discrepancy between recent MHC-I/ TAPBPR structures by examining the conformational landscape of the TAPBPR G24-R36 loop and directly probing for an interaction to the MHC-I groove using solution NMR. Our data suggest that the TAPBPR G24-R36 loop is disordered and does not form stable contacts with the floor of the empty MHC-I groove (Figs. 1d, e and 4c). These observations are consistent with the H2-D$^{d}$/TAPBPR structure in which the TAPBPR G24-R36 loop is missing due to poorly defined electron density. Our data and interpretations are also consistent with electron density for the equivalent loop of tapasin in the PLC cryo-EM structure, yet for reasons that are not clear, the authors chose to model the loop outside of the density as entering the empty MHC-I groove. One limitation of our study is that in vitro binding experiments were carried out using recombinant proteins prepared in E. coli. We note that while this is also the case for the X-ray crystallography and other biophysical studies in our field[8,16,17,29], there may be differences in MHC-I/chaperone interactions for proteins expressed in mammalian cells, primarily due to glycosylation[37]. Our 200 ns MD simulations and Rosetta modeling, performed on an HLA-A*02:01/TAPBPR complex lacking the conserved Asn86-linked glycan, suggest that the TAPBPR G24-R36 loop is highly mobile and samples conformations that do not enter the MHC-I groove (Fig. 1d, e and Supplementary Fig. 1a–e). The mobility of the tapasin loop has been investigated by recent μs-timescale MD simulations of the native peptide loading complex, which provided evidence that the Asn86-glycan moiety promotes movement of the shorter tapasin loop toward the MHC-I groove[25]. How MHC-I glycosylation at the conserved Asn86

position may alter the landscape of TAPBPR G25-R36 loop conformations remains unclear. While glycosylated MHC-I has been suggested to preferentially interact with tapasin rather than TAPBPR[37], the presence of the glycan does not inhibit the formation of a high-affinity MHC-I/TAPBPR complex expressed on CHO cells[38]. Future biophysical studies examining the role MHC-I glycosylation on chaperone interactions and peptide repertoire selection are needed to further elucidate these important aspects of MHC-I function.

Furthermore, due to the lack of an experimental HLA-A*02:01/TAPBPR structure, our study is restricted to interpreting models built from available X-ray structures as templates (Fig. 1 and Supplementary Fig. 1), although modeling is highly accurate when homology to the template is high, as is the case between MHC-I alleles. Solution NMR experiments provide clear evidence for a lack of direct interaction between the TAPBPR G24-R36 loop and the HLA-A*02:01 groove (Fig. 4c), however in vitro binding data were acquired using the human HLA-A*02:01 heavy chain and it remains unclear how differences in amino acid polymorphisms within other MHC-I, such as the mouse H2 groove, influence behavior of the TAPBPR G24-R36 loop[5,8]. While we did not find differences in the formation of empty H2-D$^d$/TAPBPR or HLA-A*02:01/TAPBPR complexes in vitro by SEC (Supplementary Fig. 4), our yeast display experiments do suggest there may be differences in the behavior of the TAPBPR G24-R36 loop as the result of allele-specific chemistry of the pMHC-I surface (Fig. 3), in agreement with recent binding results using a range of human HLA alleles[39]. Bound peptide alters the chemical features of the pMHC-I surface in an allele-dependent manner and thus both sterics and electrostatics may influence transient interactions with the intrinsically disordered loop. Future NMR experiments should probe direct interaction between the TAPBPR G24-R36 loop and empty or peptide bound MHC-I comprising different heavy chain alleles.

For measurement of $K_{D3}$ and $K_{D4}$ we obtain stoichiometry ($n$) values of 1, corresponding to the expected 1:1 stoichiometry of binding (Table 1). We observe similar n values for all peptides analyzed using TAPBPR$^{WT}$ or TAPBPR$^{\Delta G24-R36}$. In contrast, in the ITC experiments measuring $K_{D2,app}$, the resulting $n$ value is 0.4–0.5 (Table 1), suggesting that only a portion of HLA-A*02:01 is released from TAPBPR and available for binding to the titrated peptide. The remaining HLA-A*02:01 remains tightly associated with TAPBPR does not bind to peptide under the concentrations used in the ITC experiment. Another important caveat is that the heat from any dissociation of TAPBPR could contribute by decreasing or increasing the magnitude of the measured enthalpy. This effect will depend on the concentration of HLA-A*02:01/TAPBPR complex in the ITC cell, further complicating the analysis of our results. Finally, a slow off-rate of TAPBPR from HLA-A*02:01 may also contribute to limiting the concentration of MHC-I available for peptide binding. For these reasons, the ITC determined $K_{D2,app}$ represents an upper limit of the dissociation constant and, likewise, the n value is an apparent stoichiometry. We note that even though $K_{D2,app}$ reflects an upper limit, a comparison of $K_{D2,app}$ extracted from experiments using different TAPBPR constructs shows very similar values, within experimental error (Table 1). Given that peptide binding to empty MHC-I is independent of TAPBPR, our ITC data suggest no effect of the TAPBPR G24-R36 loop on binding to empty HLA-A*02:01, consistent with our NMR data (Fig. 4c).

We sought to determine the role of the G24-R36 loop on TAPBPR's recently identified dual functions of chaperoning and peptide editing. Deep mutagenesis based on the ability of tapasin or TAPBPR variants to functionally replace tapasin for HLA-A*02:01 processing within the cell suggests that the loop tip does not substantially participate in the chaperone function, while

tapasin L18 at the base of the loop makes important contacts, possibly above Y84 on the upper rim of the MHC-I groove based on the PLC cryo-EM density. However, deep mutagenesis of yeast-displayed TAPBPR indicates the G24-R36 loop does play a direct role in determining affinity with folded pMHC-I, as it occurs in peptide editing. It is well established that, in conjunction with its role as a stabilizing chaperone, TAPBPR functions as an enzyme catalyst for peptide exchange for the MHC-I[6], which requires a cycle of chaperone, peptide, and pMHC-I/MHC-I binding and unbinding events. We carefully designed novel ITC and FP assays that allowed us to uniquely probe each step of the thermodynamic cycle describing the peptide exchange process (Fig. 5a and Supplementary Fig. 6 and Supplementary Fig. 7). According to the "scoop loop" model, removal of the TAPBPR G24-R36 loop would result in a lower $K_{D4}$ (tighter binding of peptides to the MHC-I groove). Instead, we find that removal of the TAPBPR G24-R36 loop results in a higher $K_{D4}$ (Fig. 5c, g). Together, these observations, in conjunction with our loop modeling and solution NMR studies, lead us to conclude that the TAPBPR G24-R36 loop functions as a peptide trap (or lid), rather than a "scoop loop".

We have previously shown that TAPBPR acts both directly and allosterically to influence dynamics of chaperoned MHC-I molecules[29]. However, whether the TAPBPR G24-R36 loop contributes to dynamic properties of the MHC-I groove remains an open question. The lack of a stable interaction between the TAPBPR G24-R36 loop and the MHC-I groove in solution suggests that the G24-R36 loop is unlikely to influence MHC-I dynamics directly, for example, by sterically hindering movement of the MHC-I $\alpha_1/\alpha_2$ helix or $\beta$-sheet floor of the groove. We expect that, if this were the case, it would have been read out by the highly sensitive methyl resonances when comparing HLA-A*02:01/TAPBPR complexes prepared with and without the TAPBPR G24-R36 loop. Notwithstanding, our ITC, FP, and NMR data show that the TAPBPR G24-R36 loop does promote peptide binding to the empty MHC-I groove. Because the presence of bound peptide in the MHC-I regulates dynamics[29,40,41], the G24-R36 loop can indirectly contribute to MHC-I dynamics and stability through its peptide trap activity. Recent studies by X-ray crystallography of peptide-free MHC-I[24] and by solution NMR of MHC-I/TAPBPR complexes[5,29] have provided evidence of allosteric communication between the A- and F-pockets of the MHC-I. Our current NMR data suggest that, instead of participating in direct, stable interaction with the peptide or floor of the MHC-I groove, the TAPBPR G24-R36 loop promotes the formation of an intermediate state where the peptide is partially bound to the F-pocket of the MHC-I groove (Fig. 6 and Supplementary Fig. 10). Plausible models of this "trap" function involve the TAPBPR G24-R36 loop acting by either reducing the free energy barrier for peptide capture, or by reducing the dissociation of a partially bound "encounter complex". Therefore, the enhanced peptide interaction, promoted by the TAPBPR loop, may result in conformational changes at the F-pocket propagating to A-pocket residues to ultimately enable full capture of the peptide within the MHC-I groove[24].

Two recent studies investigated the putative function of the TAPBPR G24-R36 loop[20,21]. Ilca et al. examined TAPBPR constructs with a mutated but full-length G24-R36 loop, while Sagert et al. examined TAPBPR constructs with 12 or 5 residue deletions, which are similar to the TAPBPR$^{\Delta G24-R36}$ and TAPBPR$^{\Delta ALAS}$ mutants tested here. Notably, both ours and the Sagert et al. mutant TAPBPR constructs lack L30, which Ilca et al. proposed to participate in a lever-like mechanism. In corroboration with the data presented here (Figs. 4 and 5), these studies report that TAPBPR with a modified or absent G24-R36 loop can still associate with the MHC-I and that TAPBPR loop

mutants can still promote peptide exchange but in a less efficient manner. While both Ilca et al. and Sagert et al. hypothesize that interactions between the TAPBPR G24-R36 loop and the floor of the MHC-I groove are important for its function, neither study provided molecular-level data in support of this hypothesis. Here, direct observation of recombinant MHC-I proteins, albeit lacking glycosylation at Asn 86, by solution NMR provides clear evidence that the TAPBPR loop does not enter the MHC-I groove (Fig. 4 and Supplementary Fig. 5) for HLA-A*02:01, an MHC-I heavy chain allele also examined by Ilca et al. and Sagert et al. and relevant to the biology of human tapasin and TAPBPR.

In our ITC and FP assays, we explicitly control experimental conditions, such as peptide, MHC-I, and TAPBPR concentrations, to perform separate experiments under stoichiometric and substoichiometric conditions towards resolving specific steps of the peptide exchange thermodynamic cycle (Fig. 5a). Here, a major experimental parameter for distinguishing between $K_{D3}$ (association of pMHC-I with TAPBPR) and $K_{D4}$ (association of peptide with the MHC-I/TAPBPR complex) is the concentration of TAPBPR relative to the peptide and MHC-I (Supplementary Fig. 6 and Supplementary Fig. 7). This distinction allows us to deconvolute the contributions of the TAPBPR G24-R36 loop at each step of the cycle to measure apparent $K_D$ and $IC_{50}$ values. Our approach provides direct evidence for a clear function of the TAPBPR G24-R36 loop as a peptide trap (i.e., the loop decreases $K_{D4}$). In contrast, both Ilca et al. and Sagert et al. used fluorescence spectroscopy to measure peptide exchange and suggest that the TAPBPR G24-R36 loop promotes peptide dissociation from the MHC-I groove in a "scoop/lever"-like manner. The authors sought to evaluate peptide dissociation from the MHC-I/TAPBPR groove (corresponding to $K_{D4}$). In their experiments, the binding of 300 μM competitor peptide was assayed in mixtures of 1 μM TAPBPR and 300 nM MHC-I loaded with fluorescent peptide. However, when put in the context on our determined $K_D$ values, the concentrations used by Sagert et al. (3.3-fold excess compared to 20-fold excess TAPBPR used here) correspond to non-saturating TAPBPR conditions and thus do not accurately capture $K_{D4}$ values. Since our $K_D$ measurements of peptide binding to the MHC-I/TAPBPR groove are in the low to medium μM range, the concentration used by Sagert et al. was insufficient to achieve saturation of binding of the unlabeled competitor peptide to the MHC-I/TAPBPR groove. Finally, Ilca et al. performed peptide exchange/loading experiments on the surface of cells where it is unclear what concentrations of MHC-I and TAPBPR are present. These functional experiments report on a combination of events described by the processes $K_{D3}$ and $K_{D4}$, and therefore there can be a number of alternative molecular mechanisms at play that could explain the data. Our results and conclusions are consistent across different protein concentrations as probed using independent binding experiments by NMR, yeast surface display, ITC, and FP.

We further examined whether the TAPBPR G24-R36 loop may have a function in selecting peptides of a specific length distribution by FP and NMR. Our data show that the presence of the TAPBPR G24-R36 loop exerts a broad effect across the repertoire, independently of peptide length, by promoting binding of peptides according to their global affinity for the empty MHC-I groove (Supplementary Fig. 6 and Supplementary Fig. 7 and Fig. 6 and Supplementary Fig. 10), although some subtle effects were observed between TAPBPR loop mutants and specific peptide sequences by yeast display (Supplementary Fig. 3k, l). In other words, while TAPBPR lowers the peptide affinity requirements across the sampled peptidome, it appears that the specificity of peptide binding is determined by interactions with the MHC-I groove, not the TAPBPR loop.

Based on our results, we propose a model for how a longer loop in the TAPBPR sequence may contribute to shaping the displayed peptide repertoire on MHC-I molecules. The G24-R36 loop serves as a trap, allowing moderate to high-affinity peptides of lengths 8–15 from the cellular pool to associate with the MHC-I/TAPBPR complex with a binding affinity in the low micromolar range (relative to high μM, as observed for loop deletion mutants) (Fig. 7a). By lowering the affinity requirements for binding across the peptide pool, TAPBPR may function as an auxiliary peptide loader for peptides of low and moderate affinity (Fig. 7b); studies suggest that even low to moderate stability pMHC-I can provide robust T cell responses[42,43]. Local peptide concentrations will be highest in the vicinity of the TAP transporter, which acts as a hub around which PLC components associate. Tapasin lacks a long loop, but is tethered to the TAP transporter through transmembrane domain interactions[44] and therefore can load cognate peptide ligands efficiently due to their increased local concentration (Fig. 7c). In contrast, TAPBPR is not tethered to TAP, but instead utilizes a long loop acting as a trap to load peptides on empty or suboptimally loaded MHC-I molecules that have escaped the PLC (Fig. 7c). Therefore, the longer loop of TAPBPR is matched to its editing function in the Golgi where peptide concentrations will be low and only pMHC-I complexes with suboptimal peptides are to be disassembled, while tapasin has a short loop to favor competition between peptide substrates where they are in abundance surrounding the PLC (Fig. 7c). In summary, our work offers a paradigm of how two closely related molecular chaperones have developed subtle variations of both static and dynamic conformational elements around a common structural theme, to achieve unique functions that are highly adapted to their specific cellular compartments.

## Methods

**Human cell expression constructs.** For expression in human cells, cDNAs were cloned into the NheI-XhoI sites of pCEP4 (Invitrogen) with a consensus Kozak sequence. A synthetic codon-optimized sequence of human tapasin (isoform 1 a.a. 21-448; GenBank Acc. No. NP_003181) was synthesized (IDT) with a canonical signal peptide and N-terminal FLAG epitope tag (Supplementary Table 1). The cloning of TAPBPR is previously described[5]. TAPBPR-CT was constructed by PCR-based fragment assembly: the final sequence comprises a canonical signal peptide, a FLAG tag, TAPBPR residues 22-405 and tapasin residues 406-448. Both targeted mutations and generation of the libraries using oligos with degenerate NNK codons were by overlap extension PCR[27].

**Yeast expression constructs.** For yeast surface display, the extracellular domain of TAPBPR (a.a. 22-405) was cloned into the NheI-XhoI sites of pETCON[45], fusing Aga2p and a c-myc tag to the TAPBPR N- and C-termini, respectively, for surface display and detection. To generate the SSM library on the 24–35 loop, the TAPBPR insert was amplified using oligos with degenerate NNK codons by overlap extension PCR. The pooled PCR product was mixed with linearized pETCON (cut NdeI/XhoI) and electroporated in to yeast, thereby using natural homologous recombination pathways to fuse the TAPBPR insert in to the pETCON backbone.

**Human cell culture.** Expi293F cells were cultured in suspension at 37 °C, 8% $CO_2$, 125 rpm using Expi293 Expression Media (ThermoFisher)[46]. To generate the tapasin knockout line, cells were co-transfected with plasmids[46] encoding human codon-optimized Streptococcus pyogenes Cas9 and two guide RNAs targeting the tapasin gene immediately following the signal peptide (target sequences 5′-GACCCGCGG TGATCGAGTGTTGG-3′ and 5′-AACCAACACTCGATCACCGCGGG-3′). After a week, cells with reduced surface MHC-I following staining with anti-HLA-A2-PE (1/200 dilution, clone BB7.2, BioLegend Cat #343306) were enriched by sorting on a BD FACSAria II. Genomic DNA was purified (DNeasy Blood and Tissue Kit, Qiagen), and the targeted region of the tapasin gene was PCR amplified and sequenced on an Illumina MiSeq. Based on analysis with CRISPR-GA[47], >99% of tapasin gene copies in the polyclonal knockout line had indel mutations (GEO sample ID GSM3593598), whereas copies of the tapasin gene were mostly wild-type in the parental line (GEO sample ID GSM3593597). To test targeted mutants of TAPBPR or tapasin, Expi293F cells (wild-type or tapasin-KO) were transfected using ExpiFectamine (Life Technologies) with 500 ng plasmid DNA per ml of cells at a density of $2 \times 10^6$/ml. Cells were analyzed 24–26 h post-transfection. For sorting libraries, tapasin-KO Expi293F cells ($2 \times 10^6$/ml) were transfected with 1 ng

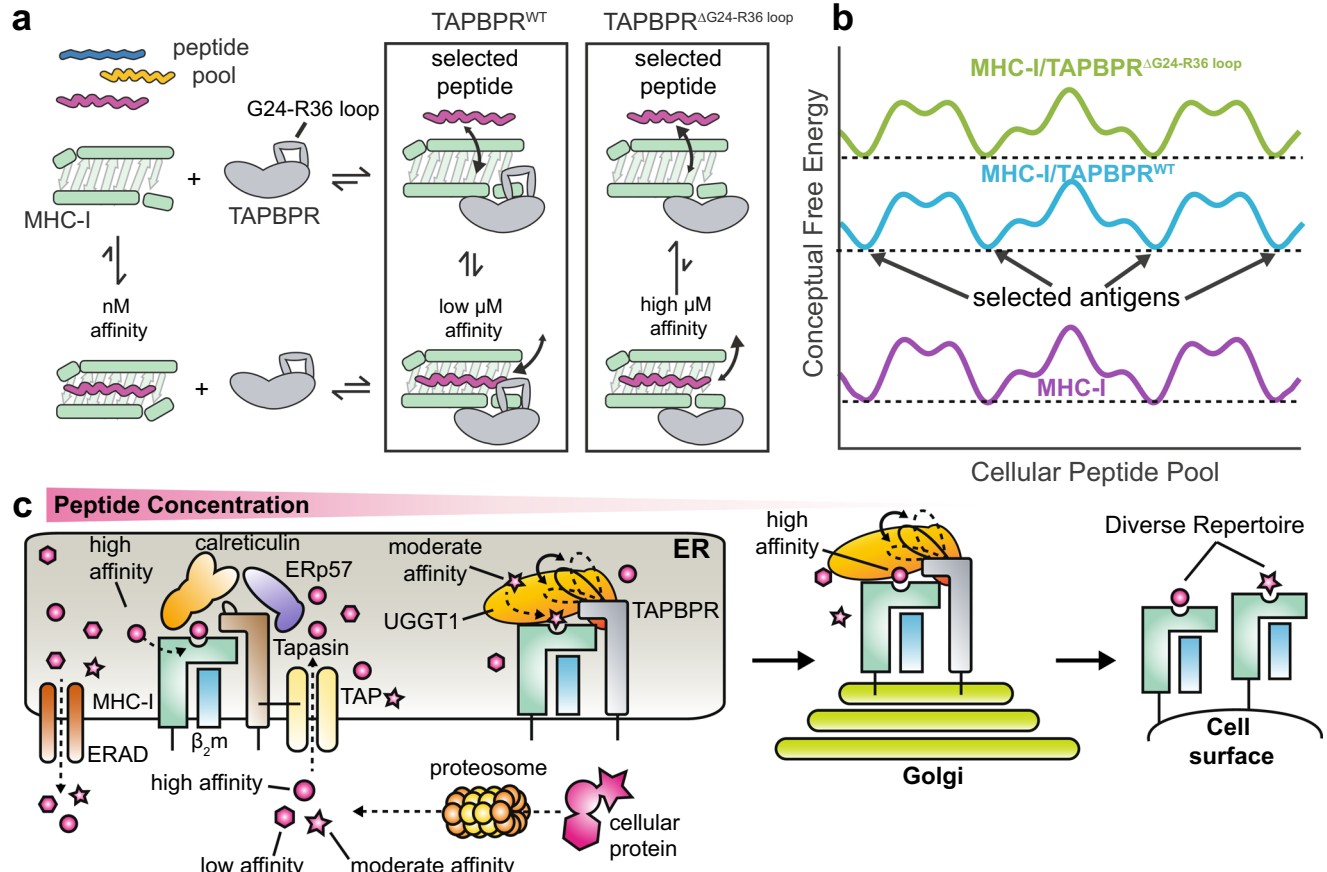

**Fig. 7 Proposed model of auxiliary peptide editing functions of TAPBPR. a** Schematic of how the TAPBPR G24-R36 loop influences the peptide exchange cycle. The G24-R36 loop serves as a trap, promoting peptide binding on the MHC-I/TAPBPR complex with low μM affinity. When TAPBPR is mutated to have a shorter loop, the affinity of peptides is reduced to high μM range. **b** Conceptual example of the free energy landscape across the cellular peptide pool. The TAPBPR G24-R36 loop shapes the selected peptide repertoire by stabilizing peptide binding across the cellular pool, functioning as an auxiliary loading chaperone for peptides of low and moderate affinity. **c** In the ER, peptide selection is governed both by the effective concentration of peptides and the absence (in tapasin) or the presence of the kinetic trap (in TAPBPR). Tapasin is tethered to the TAP transporter and because it is restricted to an environment of high peptide concentration and has a shorter loop, it primarily loads high-affinity peptides. In contrast, TAPBPR is not tethered to TAP and because it functions in environments of lower peptide concentration, it employs a trap to load both high and moderate affinity peptides and/or to minimize dissociation of peptides during transient TAPBPR/pMHC-I interactions. This process results in a diverse peptide repertoire being displayed by MHC-I molecules on the cell surface.

SSM library plasmid diluted with 1500 ng carrier plasmid (pCEP4-ΔCMV[48]) using ExpiFectamine, and the medium was replaced after 2 hours. Cells were sorted 24–25 h post-transfection.

**Flow cytometry of Expi293F cells**. To assess surface HLA-A2 expression, cells were stained on ice for 30 min with anti-HLA-A2 PE (clone BB7.2, BioLegend Cat# 343302) diluted 1:200 in PBS supplemented with 0.2% bovine serum albumin (PBS-BSA) and then washed twice with PBS-BSA. Fluorescence was measured using a BD LSRII and results were analyzed using FCS Express 6. The gating strategies for flow cytometry experiments are shown in Supplementary Fig. 11.

**Deep mutational scans of tapasin and TAPBPR-CT for chaperone activity in human cells**. Tapasin-KO Expi293F cells expressing the tapasin or TAPBPR-CT libraries were stained with diluted 1:200 anti-HLA-A2 PE (clone BB7.2, BioLegend Cat# 343302) in PBS-BSA for 30 min at 4 °C and washed twice with PBS-BSA. After gating for the main population by forward-side scatter and excluding dead cells based on DAPI uptake, the top 0.5% of cells for PE fluorescence were collected using a BD FACSAria II. RNA was extracted from sorted cells (GeneJet RNA purification kit, ThermoFisher), first-strand cDNA was synthesized with Accu-script (Agilent) primed with the EBV reverse sequencing primer, which anneals to the 3′ UTR of pCEP4-encoded transcripts. DNA fragments covering mutated regions were PCR amplified in two rounds to append sequences complementary to Illumina sequencing primers (Supplementary Table 2) and add experiment-specific barcodes and Illumina adaptamers. Products were sequenced on a NovaSeq 6000 and analyzed with Enrich[49]. Scripts for analysis are included with the GEO submission.

**Immunoblots**. Cells were pelleted, lysed in reducing SDS load dye and sonicated. Proteins were separated by electrophoresis on 10% SDS polyacrylamide gels and transferred to PVDF. For the detection of FLAG-tagged TAPBPR or tapasin, membranes were blocked with 3% BSA in Tris-buffered saline supplemented with 0.1% tween 20 (TBS-T) for 30 min, followed by staining with 1:2000 anti-FLAG (clone M2)-AP (Sigma-Aldrich Cat #A9469) for 30 min, washed five times with TBS-T, and developed using 1-Step NBT-BCIP (ThermoFisher). For the detection of cyclophilin B as a loading control, membranes were blocked with 5% nonfat milk (Biorad) in TBS-T for 30 min, followed by incubation with 1:2000 rabbit anti-cyclophilin B (Invitrogen Cat #PA1-027A) for 30 min, washed five times with TBS-T, followed by secondary incubation with 1:10,000 goat anti-rabbit HRP (Jackson ImmunoResearch Laboratories Cat# 111-035-003) for 30 min, washed five times with TBS-T, and developed using Clarity Western ECL Substrate (BioRad).

**Yeast display**. Saccharomyces cerevisiae strain EBY100 were grown in YPAD medium. To test targeted mutants, lithium acetate/polyethylene glycol 3350-treated competent yeast were heat shocked to promote plasmid uptake. For generating the TAPBPR 24–35 loop library, yeast were electroporated. Transformed yeast were selected in SDCAA medium (2% w/v glucose, 0.67% w/v yeast nitrogen base, 0.5% w/v casamino acids, 0.1 M sodium phosphate pH 6.6) at 30 °C for 1–2 days, before induction in SGCAA (in which glucose is replaced with galactose) at OD (600 nm) = 0.5 for 2 days at 24 °C. Induced EBY100 were washed with PBS-BSA and incubated with APC-conjugated MHC-I tetramers (concentrations are indicated in figure legends) and FITC-conjugated chicken anti-c-myc (1/100 dilution, Immunology Consultants Laboratory Cat# CMYC-45F) for 40 min at 24 °C, 215 rpm. Cells were washed twice with cold PBS-BSA before being resuspended for analysis on a BD Accuri C6 or sorted on a BD FACSAria II.

**Deep mutational scan of TAPBPR 24–35 loop using yeast display**. Generation of the TAPBPR library and yeast preparation are described above. Naive and sorted yeast cultures were lysed with 125 U/ml Zymolase (37 °C, 5 h) and plasmid DNA was purified using a Zymoprep kit (Zymo Research). The mutated region of TAPBPR was PCR amplified in two stages. A first round of PCR used primers that added sequences complementary for Illumina sequencing primers (Supplementary Table 2). A second round of PCR added end sequences for annealing to the Illumina flow cell and included 6 bp barcodes for unique sample identification. Amplicons were sequenced on a Illumina HiSeq 4000 and data were analyzed with Enrich[49]. Scripts for analysis are included in the GEO submission.

**Protein expression and purification**. DNA plasmid constructs encoding the luminal domain of human HLA-A*02:01 (heavy chain) and h$\beta_2$m (light chain) were generously provided by the NIH tetramer facility and transformed into BL21 (DE3) *Escherichia coli* (New England Biolabs). DNA plasmid construct encoding the luminal domain of mouse H2-D$^d$ molecule was generously provided by Kannan Natarajan, NIH. MHC-I heavy and light chain molecules were individually expressed in Luria-Broth, extracted from inclusion bodies, and refolded in vitro together with peptide at 4 °C[50]. Peptides used in this study were prepared by chemical synthesis (Biopeptik Inc, Malvern, USA or GenScript, Piscataway, USA). Peptides sequences include: P18-I10 (RGPGRAFVTI), TAX8 (LFGYPVYV), TAX9 (LLFGYPVYV), TAX10 (LLFGGYPVYV), TAX11 (LLFGGGYPVYV), TAX12 (LLFGGGGYPVYV) and KLL15 (KLLEIPDPDKNWATL). The UV-labile conditional ligands photoP18-I10 (RGPGRAF*J*TI) and photoFluM1 (KILGFVF*J*V), where *J* = 3-amino-3-(2-nitrophenyl)-propionic acid were refolded with H2-D$^d$/ h$\beta_2$m and HLA-A*02:01/h$\beta_2$m, respectively, under dark conditions[51]. For disulfide-linked peptide studies, CysTAX9 (LLFGYPVYV) and CLW15 (CLWDIETGQQKTVFV) were refolded with HLA-A*02:01 W167C, while ACW15 (ACWDIETGQQKTVFV) was refolded with HLA-A*02:01 K66C. Purification of pMHC-I complexes was performed by size-exclusion chromatography (SEC) with a HiLoad 16/600 Superdex 75 pg column at 1 mL/min with running buffer (150 mM NaCl, 25 mM Tris, pH 8). The luminal domains of various TAPBPR forms used in this study were expressed using a Drosophila S2 cell expression system in Insect-Xpress (Lanza) medium at 27 °C and induced with 1 mM CuSO$_4$ for 4 days[8]. For TAPBPR protein purification, the filtered supernatant from S2 cells was subjected to chelate agarose chromatography and bound protein was eluted using 250 mM imidazole. A second round of purification was performed by SEC on a Superdex 200 16/60 (GE Healthcare Life Science) at 1 mL/min with running buffer 1X PBS, pH 8[8]. The TAPBPR$^{\Delta\Delta LAS}$ construct was prepared by PCR using through deletion of A29-S32 using forward primer: 5′ – AAG GAC GGT GCG CAC CGT GGA AGT GAG GAC AGG GCA AGG GCC – 3′ and reverse primer: 5′ – GGC CCT TGC CCT GTC CTC ACT TCC ACG GTG CGC ACC GTC CTT – 3′, and confirmed by DNA sequencing. The TAPBPR$^{WT}$ and TAPBPR$^{\Delta G24\text{-}R36}$ constructs were generously provided by Kannan Natarajan, NIH. DNA transfection of the TAPBPR constructs into Drosophila S2 cells was performed using standard protocols with X-tremeGENE 9 (Sigma-Aldrich). Following expression and purification all proteins were exhaustively buffer exchanged into 50 mM NaCl, 20 mM sodium phosphate pH 7.2.

**Sequence alignment**. Sequence alignment was performed between *Homo sapiens* (Hs, human) TAPBPR (UniProtID: Q9BX59) and tapasin (UniProtID: O15533), *Mus musculus* (Mm, mouse) TAPBPR (UniProtID: Q8VD31) and tapasin (Uni-ProtID: Q9R233), and *Rattus norvegicus* (Rn, rat) TAPBPR (UniProtID: D4A6L1) and tapasin (UniProtID: Q99JC6) using ClustalOmega.

**Disulfide design constructs**. Disulfide linked constructs for CysTAX9, ACW15 (ACWDIETGQQKTVFV), and CLW15 (CLWDIETGQQKTVFV) were designed using Disulfide by Design v2 (http://cptweb.cpt.wayne.edu/DbD2/)[52] using PDB ID 1DUZ or a RosettaCM model based on PDB ID 4U6Y.

**Rosetta modeling**. Rosetta modeling of MHC-I/TAPBPR complexes and the TAPBPR G24-R36 loop was performed using either RosettaCCD[53], RosettaKIC[54], or RosettaCM[55] using template X-ray structure PDB ID 5WER or PDB ID 5OPI. In each case the sequence for H2-D$^d$ or H2-D$^b$ was replaced with HLA-A*02:01 using Rosetta's partial_thread application. Fragment files for loop modeling were generated using the Robetta server (http://robetta.bakerlab.org/). The lowest energy structures were selected from a total of 1000 models calculated for each protocol. Rosetta modeling of TAX10, TAX11 or TAX12 peptides in complex with HLA-A*02:01 was performed using RosettaCM against PDB ID 1HHH, 5D9S, and 4JQX, respectively.

**MD simulations**. All-atom MD simulations in explicit solvent were carried out in GROMACS version 2019.2 using an AMBER99SB-ILDN protein force field and TPI3P water model[29]. The input structure was the lowest energy RosettaCM model of peptide-deficient HLA-A*02:01/h$\beta_2$m/TAPBPR built from template PDB ID 5OPI. LINCS and SETTLE constraint algorithms were used to constrain protein and water molecules, respectively. Virtual site hydrogens were used to allow an integration time step of 4 fs[56]. Coordinates were output every 10 ps. Short-range interactions were treated with a Verlet cut-off scheme with 10 Å electrostatic and

van der Walls cutoffs and long-range electrostatics were treated with the PME method with a grid spacing of 1.2 Å and cubic interpolation. Periodic dodecahedron boundaries were used. The thermodynamic ensemble was nPT where the temperature was kept constant at 300 K by a V-rescale modified thermostat with 0.1 ps time constant and pressure was kept constant at 1 bar pressure using an isotropic Berendsen barostat. The system was solvated to overall neutral charge and contained Na$^+$ and Cl$^-$ ions to yield physiological concentration of 0.15 M. Following 500 steps of steepest-descent energy minimization, initial velocities were generated at 65 K with linear heating up to 300 K over 2 ns. Trajectories were acquired for 200 ns. Structures were extracted after the final 200 ns simulations using GROMACS.

**Differential scanning fluorimetry**. DSF experiments were performed on an Applied Biosystems ViiA 7 qPCR machine with excitation and emission wavelengths set to 470 nm and 569 nm with proteins in buffer of 50 mM NaCl, 20 mM sodium phosphate pH 7.2. Experiments were conducted in triplicate in MicroAmp Fast 96-well plates with 50 μL total volume containing final concentrations of 7 μM protein and 10× SYPRO orange dye (ThermoFisher). The temperature was incrementally increased at a scan rate of 1 °C/min between 25 °C and 95 °C. Data analysis and fitting were performed in GraphPad Prism v7.

**Circular dichroism**. Far-UV CD spectra were acquired using a JASCO J-815 Spectropolarimeter. CD spectra of wild-type TAPBPR, TAPBPR$^{\Delta\Delta LAS}$, and TAPBPR$^{\Delta G24\text{-}R36}$ were acquired using 0.05 mg/mL protein in 2 mL of 50 mM NaCl, 20 mM sodium phosphate pH 7.2 in a quartz cuvette. A buffer blank was acquired and subtracted from each CD spectra. CD spectra were acquired from 190 to 260 nm in triplicate at 25 °C with a scan rate of 50 nm/min. The experimental CD values of ellipticity (mdeg) were converted to molar ellipticity ($\theta$ = deg cm$^2$/dmol).

**NMR spectroscopy**. For NMR performed on labeled heavy chain, samples were prepared using either ILV$^{proS}$ (Ile $^{13}$C$\delta$1, Leu $^{13}$C$\delta$2, Val $^{13}$C$\gamma$2) or AILV methyl (Ala $^{13}$C$\beta$, Ile $^{13}$C$\delta$1, Leu $^{13}$C$\delta$1/$^{13}$C$\delta$2, Val $^{13}$C$\gamma$1/$^{13}$C$\gamma$2) isotopic labeling at the HLA-A*02:01 heavy chain against a $^{12}$C/$^2$H/$^{15}$N background. Bound peptide, h$\beta_2$m or TAPBPR were fully protonated. NMR methyl resonance assignments of free and TAPBPR bound states of HLA-A*02:01 were reported previously by our group[5]. Peptide-deficient HLA-A*02:01/h$\beta_2$m/TAPBPR complexes were prepared as described above in the section "Preparation of empty MHC-I/TAPBPR complexes" where HLA-A*02:01 was ILV$^{proS}$ or AILV methyl labeled. The resulting purified HLA-A*02:01/h$\beta_2$m/TAPBPR$^{WT}$, HLA-A*02:01/h$\beta_2$m/TAPBPR$^{\Delta\Delta LAS}$, or HLA-A*02:01/h$\beta_2$m/TAPBPR$^{\Delta G24\text{-}R36}$ complexes were exhaustively dialyzed into NMR buffer (50 mM NaCl, 20 mM sodium phosphate pH 7.2, 5% D$_2$O) and concentrated to ~ 80 μM. For NMR performed on labeled TAX9, CysTAX9 or CLW15 peptides, samples were $^{15}$N/$^{13}$C LV or ILV isotopically labeled against a protonated HLA-A*02:01 and h$\beta_2$m background. $^{15}$N/$^{13}$C ILV or LV isotopically labeled peptides were obtained from GenScript (Piscataway, USA). Free peptide, peptide in complex with HLA-A*02:01/h$\beta_2$m or peptide/HLA-A*02:01/h$\beta_2$m in the presence of 8-fold molar excess TAPBPR were exhaustively dialyzed into NMR buffer (50 mM NaCl, 20 mM sodium phosphate pH 7.2, 5% D$_2$O). Spectra were obtained at ~100 μM free peptide and pMHC-I concentrations. Two-dimensional $^1$H-$^{13}$C methyl SOFAST HMQC experiments[57] were recorded at 25 °C at a $^1$H field strength of 800 or 600 MHz. A total number of 320 scans were used with a 0.2 s recycle delay (d1) and acquisition times of 12 and 30 ms in the $^{13}$C dimension for labeled heavy chain and labeled peptide, respectively. Data were processed with 4 and 10 Hz Lorentzian line broadening in the direct and indirect dimensions. Chemical shift deviations (CSD, p.p.m.) were determined between TAX9/HLA-A*02:01/h$\beta_2$m and TAPBPR bound HLA-A*02:01/h$\beta_2$m using the equation $\Delta\delta^{CH3} = [1/2(\Delta\delta_H{}^2 + \Delta\delta_C{}^2/4)]^{1/2}$ for each methyl resonance. All NMR data were recorded using TopSpin 3.5pl7 (Bruker), processed with NMRPipe[58] and analyzed using NMRFAM-SPARKY[59].

**Isothermal titration calorimetry**. ITC was performed using a MicroCal VP-ITC system (Malvern Panalytical, Westborough, MA). All proteins were exhaustively dialyzed into the buffer (50 mM NaCl, 20 mM sodium phosphate pH 7.2) filtered through a 0.22 μm PES membrane. ITC experiments to probe the $K_{D2}$ step of the peptide exchange cycle were performed in the absence of excess TAPBPR to allow for dissociation of HLA-A*02:01 from TAPBPR in the presence of incoming peptide. Syringe containing ~50–100 μM peptide was titrated into a calorimetry cell containing ~15 μM purified peptide-deficient HLA-A*02:01/h$\beta_2$m/TAPBPR complex. ITC experiments to probe the $K_{D3}$ step of the peptide exchange cycle were performed by titration of ~150–200 μM purified pMHC-I into a calorimetry cell containing ~15 μM TAPBPR and 1 mM excess peptide. ITC experiments to probe the $K_{D4}$ step of the peptide exchange cycle were performed under stoichiometric conditions in the presence of excess TAPBPR to minimize dissociation of TAPBPR from the pMHC-I/TAPBPR complex. Syringe containing ~150–200 μM peptide was titrated into a calorimetry cell containing ~15 μM peptide-deficient HLA-A*02:01/h$\beta_2$m/TAPBPR complex and 50 μM excess TAPBPR. In all ITC experiments injection volumes were 10 μL performed for a duration of 10 s and spaced 220 s apart to allow for a complete return to baseline. Data were subtracted from a control experiments. Data were processed and analyzed with Origin software.

Isotherms were fit using a one-site ITC binding model. The first data point was excluded from analysis. Each ITC experiment was performed with one technical replicate. Error bars were determined from fitting to a one-site binding model. Determined $K_D$ and n (stoichiometry) values are noted in Table 1.

**Fluorescence polarization.** FP was performed using a modified TAX9 peptide labeled with fluorescent TAMRA dye ($K^{TAMRA}$LFGYPVYV, herein called TAMRA-TAX9) (Biopeptik Inc, Malvern, USA). Peptide-deficient HLA-A*02:01/$h\beta_2m$/TAPBPR complexes for FP were prepared by UV-irradiation at 365 nm for 1 h of photoFluM1/HLA-A*02:01/$h\beta_2m$/TAPBPR complexes followed by purification, as described in the section "Preparation of empty MHC-I/TAPBPR complexes". FP experiments to probe the $K_{D2}$ step of the peptide exchange cycle (defined as $IC_{50\ 2}$ in the FP competition experiments) were performed under substoichiometric conditions in the absence of excess TAPBPR to allow for dissociation of HLA-A*02:01/$h\beta_2m$ from TAPBPR in the presence of incoming peptide. Graded concentrations (0, 2.5, 5, 10, 25, 50, 100, 500, 1000, 2000, 3000, 4000 and 5000 nM) of TAX8, TAX9, TAX10, TAX11, TAX12, or KLL15 were added to a mixture of 1 nM TAMRA-TAX9 and either 50 nM of peptide-deficient HLA-A*02:01/$h\beta_2m$/TAPBPR$^{WT}$, peptide-deficient HLA-A*02:01/$h\beta_2m$/TAPB-PR$^{\Delta ALAS}$, or peptide-deficient HLA-A*02:01/$h\beta_2m$/TAPBPR$^{\Delta G24-R36}$. FP experiments to probe the $K_{D3}$ step of the peptide exchange cycle were performed by titration of graded concentrations (0, 0.1, 2, 4, 10, 30, 50, and 80 μM) of TAPBPR$^{WT}$, TAPBPR$^{\Delta ALAS}$ or TAPBPR$^{\Delta G24-R36}$ into 1 nM TAMRA-TAX9 and 50 nM TAX8/HLA-A*02:01/$h\beta_2m$. FP experiments to probe the $K_{D4}$ step of the peptide exchange cycle (defined as $IC_{50\ 4}$ in the FP competition experiments) were performed under stoichiometric conditions in the presence of excess TAPBPR$^{WT}$, TAPBPR$^{\Delta ALAS}$ or TAPBPR$^{\Delta G24-R36}$ to minimize dissociation of TAPBPR from the pMHC-I/TAPBPR complex. Graded concentrations (0, 10, 50, 250, 500, 1000, 2000, 3000, 4000, 6000, 8000, 10,000, 50,000, 100,000 nM) of TAX8, TAX9, TAX10, TAX11, TAX12, or KLL15 were added to a mixture of 1 nM TAMRA-TAX9 and either 50 nM of peptide-deficient HLA-A*02:01/$h\beta_2m$/TAPBPR$^{WT}$, peptide-deficient HLA-A*02:01/$h\beta_2m$/TAPBPR$^{\Delta ALAS}$, or peptide-deficient HLA-A*02:01/$h\beta_2m$/TAPBPR$^{\Delta G24-R36}$ together with 1 μM of their respective free TAPBPR$^{WT}$, TAPBPR$^{\Delta ALAS}$ or TAPBPR$^{\Delta G24-R36}$. Each experiment was performed in a volume of 140 μL and loaded onto a black 96-well polystyrene assay plate (Costar 3915). FA data were recorded via a Perkin-Elmer Envision 2103 plate reader with excitation filter $\lambda_{ex} = 531$ nm and emission filter $\lambda_{em} = 595$ nm with measurement height 4.3, excitation light 100, G-factor 1.36, and a total of 100 flashes. In each of the above experiments, the average of FP after incubation for 95-105 minutes 25 °C was plotted as a function of the $\log_{10}$ of excess peptide. Each experiment was performed in triplicate and is representative of at least two independent experiments. Experimental values were subtracted from background FA values obtained from incubation of TAMRA-TAX9 alone. All samples were prepared in matched buffer (50 mM NaCl, 20 mM sodium phosphate pH 7.2, 0.05% (v/v) tween-20). Data were fit using GraphPad Prism v7.

**Reporting summary**. Further information on experimental design is available in the Nature Research Reporting Summary linked to this paper.

## Data availability
Plasmids are deposited with Addgene (ID numbers 141308-9 and 153471-8). All Illumina sequencing data is deposited with GEO under series accession numbers GSE147137, GSE126206, GSE159247, and GSE118568. NMR assignments have been deposited into the Biological Magnetic Resonance Data Bank (http://www.bmrb.wisc.edu) under accession numbers 28107 and 28108. Previously solved structures were obtained from the Protein Data Bank (https://www.rcsb.org/). Other data are available from the corresponding authors upon reasonable request. Source data are provided with this paper.

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

## Acknowledgements

We are grateful to Kannan Natarajan and David Margulies (NIH) for helpful comments and for providing the insect cell lines and DNA constructs for TAPBPR[WT] and TAPBPR[ΔG24-R36] protein expression. We thank Arne Schön (Johns Hopkins) for helpful comments on ITC experiments. E.P. was supported through NIAID (R01AI143997). N. G.S. was supported through NIAID (5R01AI143997), NIGMS (5R35GM125034) and High-End Instrumentation (HIE) Grant S10OD018455, which funded the 800 MHz NMR spectrometer at UCSC. We acknowledge the use of the Fox Chase Cancer Center NMR facility. Flow cytometry and Illumina sequencing were supported by the UIUC Roy J. Carver Biotechnology Center.

## Author contributions

A.C.M., C.A.D., E.P., and N.G.S. designed research. A.C.M., C.A.D., N.A., and E.P. performed research. G.I.M., S.A.O., D.M., and N.A. contributed reagents/analytic tools. A.C.M., C.A.D., E.P., and N.G.S. wrote the manuscript.

## Competing interests

E.P. is a cofounder of Orthogonal Biologics, Inc; the company had no role in this study. N.G.S. is a cofounder of MultiplexThera, Inc., and a named inventor on licensed patents concerning the preparation of peptide-receptive MHC-I molecules; the company had no role in this study. The other authors have no competing interests.
