## [Peer Review File · Nature Communications]

REVIEWER COMMENTS

Reviewer #1 (Remarks to the Author):

The manuscript by McShan et al. describes a biophysical study of the G24-R36 loop in the TAPBPR chaperone for MHC-I molecules. Previous work suggested that this loop played a role in peptide editing by stabilizing the floor of the empty MHC groove and acting as a “scoop loop”. However, the present study suggests that the loop does not form direct stable interactions with MHC-I, but rather acts as a trapping mechanism for incoming peptides, thereby lowering inherent peptide affinity requirements. The work is rigorous, employing a wide range of biophysical tools including solution NMR spectroscopy, CD, ITC, and fluorescence polarization as well as extensive mutational studies. Overall, the results from the study help to explain a number of contradictions in previous structural reports and provides a working model for how the TAPBPR and tapasin chaperones function in the ER.

My specific comments relate to the NMR portion of the manuscript:

1. What is the molecular weight of the TAPBPR/MHC complex? I don't see it mentioned but it is particularly relevant to the NMR study.
2. In Fig. 4C, the spectrum of the MHC/wt-TAPBPR complex is similar to the spectrum with TAPBPR loop deletion mutants, implying that there are no stable interactions of the G24-R36 loop with MHC. But an overlay of the spectra and/or a plot of CSD (as in Fig S5) would help to show this better. Also, what about peak intensity changes? It looks like there may be some (e.g. V95, I124) but the authors don't mention it. Can the authors comment on the possible role of the TAPBPR loop in MHC dynamics?
3. Are there other methyl probes in the MHC that are near the interface with TAPBPR (but not the loop) that do shift? If so, it would be good to show these as a control (evidence of binding). Do they match the epitope from the X-ray and cryoEM studies?

Minor:

4. Dotted boxes in Fig 4C are hard to see; change to a darker color.
5. Line 465 should be F-pocket.

Reviewer #2 (Remarks to the Author):

The submitted manuscript proposes a substrate gating mechanism (referred to as a “peptide trap”) for the G24-R36 loop of TAPBPR. The authors use a wide variety of experimental and computational techniques to conclude that the loop is not involved in the MHC-I chaperone activity of TAPBPR, but rather captures incoming peptides and stabilises partially bound ones, thus contributing to peptide edition.

These results are significant since they help resolve the recent controversy regarding the putative role of the G24-R36 loop as a “peptide scoop”, and generally improve our understanding of how peptide editors function to shape the immune response. The results are somewhat limited in scope, however, since, in the main, they focus on a single catalytically relevant loop of a single peptide editor.

The authors' multidisciplinary approach encompasses in vivo essays, enzymology, biophysical experimental techniques including NMR spectroscopy, and theoretical techniques such as molecular dynamics (MD) simulations. This allows them to draw very solid conclusions that are usually supported by several techniques. At the same time, the

authors do not neglect the details of any specific technique as far as I can tell from my personal expertise. In particular, the authors are to be commended for providing an MD simulation protocol that is clear and detailed enough to be easily reproducible.

The literature review is generally thorough. The analysis of TAPBPR structures and the PLC electron density, for instance, adequately questions why the scoop loop hypothesis was proposed in the first place.

Major issues:

200 ns of MD simulation is much too short to conclude that the G24-R36 loop does not interact with the floor of the MHC-I binding groove because loop dynamics often proceed on the μ s timescale. By comparison, recent MD simulations of the full PLC (Fisette et al., 2020) show the corresponding loop in tapasin interacting with the rim of the groove (L18-Y84), but only two such events are observed over 5 μ s. Much longer simulations must be provided to support the authors' claims. This does not invalidate the authors' general conclusions, however, since they are also supported by NMR spectroscopy results. In any case, these MD results should be compared to those from the aforementioned computational study.

There is no validation whatsoever of MD simulations in the manuscript. At the very least, global RMSD time series should be included in supporting information to assess if the protein complex and its individual subunits are stable and if the system has reached equilibrium.

Minor points:

p. 4: "while applying standard MD constraints over the entire system" This is an odd formulation. Considering that sufficient details are given in Methods, this could be left out.

p. 5: "We hypothesize that TAPBPR-CT may be recruited in to the PLC." Considering the architecture of the PLC, this is very unlikely; how would ERp57 and Crt fit in such a chimera? And if they are absent, can this still be called the PLC? Probably the authors meant that TAPBPR-CT recruits TAP thanks to the tapasin TM and cytosolic domains. This ought to be clarified.

Recent results by the Springer group (Anjanappa et al., 2020) that show synergistic (A-pocket, F-pocket) peptide-MHC-I binding should be discussed in the context of the peptide trap "early capture" mechanism proposed here.

The constraints (LINCS and SETTLE) used in the MD simulations correspond to a 2-fs integrator, but the authors used 4 fs. Probably, they forgot to mention that they also use virtual sites for hydrogens to allow for 4 fs. Otherwise, their simulations would presumably not have been stable.

--

Olivier Fisette
Advanced Research Computing, ICT
University of Saskatchewan, Canada

Reviewer #3 (Remarks to the Author):

This is a strong and compelling manuscript that addresses the mechanism of tapasin/TAPBPR in peptide loading and exchange. In exquisite detail, the authors have addressed the mechanism of peptide selection and addressed complexities such as the role of peptide properties and the impact of MHC variants. The manuscript is a physical tour de force and provides an excellent example of using fundamental protein biophysics to address challenging immunological questions.

I have one concern that needs to be addressed. The ITC and FP experiments purport to get at binding of peptide to empty MHC-I proteins. To get at this the authors stabilize "empties" using substoichiometric TAPBPR. Stoichiometry notwithstanding, their measurements will still be influenced by the enthalpy of TAPBPR dissociation during the titration. While throughout the text they are careful to use the phrase "apparent K_d ", and they have some language elsewhere that alludes to the fact they are not exactly measuring peptide binding to empties, this point is lost in presentation of Fig. 5A and the discussion around it. Given the rigor elsewhere in the paper, the authors should be more clear about this. While the values obtained are realistic as the authors point out, increasing clarity is important and will not decrease the value or conclusions of the paper.

Similarly, the authors do not indicate the n values from the ITC experiments. These are necessary to evaluate the quality of the K_d values determined by ITC data, as well as the models chosen. For K_{d2} , it could possibly speak to the impact of TAPBPR dissociation in contributing to heats. As a final point, the concentration of substoichiometric TAPBPR would decrease as sample is injected, further complicating exactly what is measured - this should be clarified, and possibly could be addressed via the n values as indicated above.

A minor point: what was the rationale for the single mutants that were selected in the targeted mutations in the G24-R36 loop (lines 169-170 on p. 4)? Although the more comprehensive mutational experiments follow, there should be an explanation of why these particular mutations were chosen.

Reviewer #1 (Remarks to the Author):

The manuscript by McShan et al. describes a biophysical study of the G24-R36 loop in the TAPBPR chaperone for MHC-I molecules. Previous work suggested that this loop played a role in peptide editing by stabilizing the floor of the empty MHC groove and acting as a “scoop loop”. However, the present study suggests that the loop does not form direct stable interactions with MHC-I, but rather acts as a trapping mechanism for incoming peptides, thereby lowering inherent peptide affinity requirements. The work is rigorous, employing a wide range of biophysical tools including solution NMR spectroscopy, CD, ITC, and fluorescence polarization as well as extensive mutational studies. Overall, the results from the study help to explain a number of contradictions in previous structural reports and provides a working model for how the TAPBPR and tapasin chaperones function in the ER.

We thank Reviewer #1 for their positive and constructive comments. In the revised manuscript, we have provided additional NMR analysis to help clarify and strengthen our conclusions.

My specific comments relate to the NMR portion of the manuscript:

1. What is the molecular weight of the TAPBPR/MHC complex? I don't see it mentioned but it is particularly relevant to the NMR study.

The molecular weight of the MHC-I/TAPBPR complex is 87 kDa. We agree with the reviewer that this information is relevant to our NMR study because the size of the complex prompted the use of perdeuteration and selective methyl labelling. We have added to this information to the revised manuscript in lines 299-302.

2. In Fig. 4C, the spectrum of the MHC/wt-TAPBPR complex is similar to the spectrum with TAPBPR loop deletion mutants, implying that there are no stable interactions of the G24-R36 loop with MHC. But an overlay of the spectra and/or a plot of CSD (as in Fig S5) would help to show this better.

In the revised manuscript, we modified Figure 4 in the following ways:

First, in Figure 4C we now provide an overlay of the full 2D ^1H - ^{13}C methyl HMQC spectra of HLA-A*02:01/TAPBPR complexes prepared with TAPBPR^{WT} versus those formed with TAPBPR ^{Δ ALAS} and TAPBPR ^{Δ G24-R36}. Because it is difficult to visualize full spectra overlays, we also include detailed overlays focusing on specific HLA-A*02:01 methyl-bearing residues at the interface with the TAPBPR G24-R36 loop in Figure S5.

Second, in the revised Figure 4D we have added a chemical shift deviation (CSD) analysis of HLA-A*02:01/TAPBPR complexes formed using either TAPBPR^{WT} or TAPBPR ^{Δ G24-R36} (relative to unbound TAX9/HLA-A*02:01). In agreement with our previous conclusions, the new analysis highlights that the CSD profiles for formation of the HLA-A*02:01/TAPBPR complex are not influenced by the presence of the TAPBPR G24-R36 loop.

The new CSD analysis is included in the revised main text **in lines 317-326**.

Also, what about peak intensity changes? It looks like there may be some (e.g. V95, I124) but the authors don't mention it.

We apologize that in the initial version of Figure 4C the 2D ^1H - ^{13}C methyl HMQC spectra of the HLA-A*02:01/TAPBPR complexes were not contoured at exactly the same level, and so it may have seemed like there were significant peak intensity changes. This has been corrected in the updated manuscript.

To show this explicitly, we have included in the new Figure 4D an intensity ratio analysis that reveals that the methyl groups affected upon complex formation with TAPBPR are not influenced by the presence of the TAPBPR G24-R36 loop, across the entire HLA-A*02:01 molecule.

Finally, the intensity of specific HLA-A*02:01 methyl residues at the interface with the TAPBPR loop can be compared in excerpts presented in Figure S5.

The new intensity analysis is included in the revised main text **in lines 317-326**.

Can the authors comment on the possible role of the TAPBPR loop in MHC dynamics?

We have previously shown that TAPBPR acts both directly and allosterically to influence dynamics of chaperoned MHC-I molecules (McShan et al., 2018). However, whether the TAPBPR G24-R36 loop contributes to dynamic properties of the MHC-I groove remains an open question. The lack of a stable interaction between the TAPBPR G24-R36 loop and the MHC-I groove in solution suggests that the G24-R36 loop is unlikely to influence MHC-I dynamics directly, for example, by sterically hindering movement of the MHC-I α_1/α_2 helix or β -sheet floor of the groove. We expect that, if this was the case, it would have been read out by the highly sensitive methyl resonances when comparing HLA-A*02:01/TAPBPR complexes prepared with and without the TAPBPR G24-R36 loop.

Notwithstanding, our ITC, FP and NMR data show that the TAPBPR G24-R36 loop does promote peptide binding to the empty MHC-I groove. Because the presence of bound peptide in the MHC-I regulates dynamics (Ayres et al., 2019; Hein et al., 2014; McShan et al., 2018), the G24-R36 loop can indirectly contribute to MHC-I dynamics and stability through its peptide trap activity. Recent studies by X-ray crystallography of peptide-free MHC-I (Anjanappa et al., 2020) and by solution NMR of MHC-I/TAPBPR complexes (McShan et al., 2018, 2019) have provided evidence of allosteric communication between the A- and F-pockets of the MHC-I. Our current NMR data suggest that, instead of participating in a direct, stable interaction with the peptide or floor of the MHC-I groove, the TAPBPR G24-R36 loop promotes the formation of an intermediate state where the peptide is partially bound to the F-pocket of the MHC-I groove (Fig. 6, Fig. S10). Plausible models of this “trap” function involve the TAPBPR G24-R36 loop acting by either reducing the free energy barrier for peptide capture, or by reducing the dissociation of a partially bound “encounter complex”.

Therefore, the enhanced peptide interaction, promoted by the TAPBPR loop, may result in conformational changes at the F-pocket propagating to A-pocket residues to ultimately enable full capture of the peptide within the MHC-I groove (Anjanappa et al., 2020).

We have included this discussion in revised text in lines 596-619.

3. Are there other methyl probes in the MHC that are near the interface with TAPBPR (but not the loop) that do shift? If so, it would be good to show these as a control (evidence of binding). Do they match the epitope from the X-ray and cryoEM studies?

Yes, there are methyl probes along HLA-A*02:01, near the identified X-ray/cryoEM interface for the homologous MHC-I/TAPBPR interactions, that shift upon complex formation with TAPBPR. Prompted by the reviewer's comments, we believe the revised manuscript would benefit from a direct comparison of chemical shift changes on models of the HLA-A*02:01/TAPBPR complexes with and without the TAPBPR G24-R36 loop. We now present this analysis in the new Figure 4D, E. The analysis reveals that the binding interface between HLA-A*02:01 and TAPBPR is not influenced by the TAPBPR G24-R36 loop. Furthermore, our solution NMR experiments corroborate the MHC-I/chaperone interaction surfaces identified by the previous NMR, X-ray and cryoEM studies. We have incorporated this information in revised main text in lines 317-326.

Minor:

4. Dotted boxes in Fig 4C are hard to see; change to a darker color.

We have changed the color of the dotted boxes to dark gray in the revised Figure 4C. We hope this is easier to visualize.

5. Line 465 should be F-pocket.

Fixed.

Reviewer #2 (Remarks to the Author):

The submitted manuscript proposes a substrate gating mechanism (referred to as a “peptide trap”) for the G24-R36 loop of TAPBPR. The authors use a wide variety of experimental and computational techniques to conclude that the loop is not involved in the MHC-I chaperone activity of TAPBPR, but rather captures incoming peptides and stabilises partially bound ones, thus contributing to peptide editing.

These results are significant since they help resolve the recent controversy regarding the putative role of the G24-R36 loop as a “peptide scoop”, and generally improve our understanding of how peptide editors function to shape the immune response. The results are somewhat limited in scope, however, since, in the main, they focus on a single catalytically relevant loop of a single peptide editor.

The authors’ multidisciplinary approach encompasses in vivo essays, enzymology, biophysical experimental techniques including NMR spectroscopy, and theoretical techniques such as molecular dynamics (MD) simulations. This allows them to draw very solid conclusions that are usually supported by several techniques. At the same time, the authors do not neglect the details of any specific technique as far as I can tell from my personal expertise. In particular, the authors are to be commended for providing an MD simulation protocol that is clear and detailed enough to be easily reproducible.

The literature review is generally thorough. The analysis of TAPBPR structures and the PLC electron density, for instance, adequately questions why the scoop loop hypothesis was proposed in the first place.

We thank Reviewer #2 for their kind comments and constructive criticism, particularly on our MD simulations. Below we provide a point-by-point response to the concerns raised by the reviewer.

Major issues:

200 ns of MD simulation is much too short to conclude that the G24-R36 loop does not interact with the floor of the MHC-I binding groove because loop dynamics often proceed on the μ s timescale. By comparison, recent MD simulations of the full PLC (Fisette et al., 2020) show the corresponding loop in tapasin interacting with the rim of the groove (L18-Y84), but only two such events are observed over 5 μ s. Much longer simulations must be provided to support the authors’ claims. This does not invalidate the authors’ general conclusions, however, since they are also supported by NMR spectroscopy results. In any case, these MD results should be compared to those from the aforementioned computational study.

Protein loops may exhibit changes on different timescales, depending on their length and amino acid composition. These timescales range from fast (<100 ns) to intermediate (~500 ns) to slow (>1 μ s) (Gu et al. J. Chem. Theory Comput. 2015). The MD simulations performed by Fisette et al. PNAS. 2020 do indeed suggest that the shorter tapasin loop moves on the intermediate to slow timescale.

While we agree that the timescale of our MD simulation is relatively short, our aim is to examine how stable a similar conformation to the published TAPBPR “scoop loop” was in our HLA-A*02:01/TAPBPR complex, and not to perform a detailed free energy/PMF calculation. Even within the 200 ns simulation timescale, we do observe movement of the TAPBPR G24-R36 loop away from the MHC-I groove (Figure 1D, E). We agree with the reviewer that due to the limited simulation timescale, our MD results are more suggestive rather than providing conclusive evidence. However, as mentioned by the reviewer, the MD simulation is just one piece of evidence in our manuscript corroborating the lack of strong interaction between the TAPBPR G24-R36 loop and the MHC-I groove.

We have included an additional sentence to make the caveats of our MD simulations more transparent in lines 158-164 of the revised manuscript, and in the revised discussion, we have also included a more detailed comparison with the work of Fiset et al. PNAS. 2020 in lines 525-538.

There is no validation whatsoever of MD simulations in the manuscript. At the very least, global RMSD time series should be included in supporting information to assess if the protein complex and its individual subunits are stable and if the system has reached equilibrium.

As per the reviewer’s suggestion, in the revised Figure S1 we have provided a new plot showing C_α root-mean-square deviation (RMSD) from the starting structure as a function of simulation time. RMSD has been calculated both for the entire MHC-I/TAPBPR complex as well as for the individual components of the system, relative to a global reference frame. The MHC-I/TAPBPR complex and its components exhibit C_α RMSD values that are in a similar range to those observed by other groups performing MD simulations on analogous systems (Fiset et al. Sci Rep. 2016; Fiset et al. PNAS. 2020; Anjanapp et al. Nature Communications. 2020). The plot verifies the global stability of the entire complex and its individual subunits within our simulation timescale, and shows that the system has reached equilibrium. We have included this new analysis in lines 141-148 of the revised main text.

Minor points:

p. 4: “while applying standard MD constraints over the entire system” This is an odd formulation. Considering that sufficient details are given in Methods, this could be left out.

Prompted by the reviewers comment, we have removed the phrase in revised manuscript.

p. 5: “We hypothesize that TAPBPR-CT may be recruited in to the PLC.” Considering the architecture of the PLC, this is very unlikely; how would ERp57 and Crt fit in such a chimera? And if they are absent, can this still be called the PLC? Probably the authors meant that TAPBBR-CT recruits TAP thanks to the tapasin TM and cytosolic domains. This ought to be clarified.

The reviewer is correct; we hypothesize that the tapasin TM domain will recruit TAP, but did not intend to imply recruitment of other PLC components such as ERp57 and Crt. We have modified the sentence as follows (lines 207-208):

“We hypothesize that TAPBPR-CT may be recruited to the TAP transporter via the tapasin C-terminal region...”

Recent results by the Springer group (Anjanappa et al., 2020) that show synergistic (A-pocket, F-pocket) peptide-MHC-I binding should be discussed in the context of the peptide trap “early capture” mechanism proposed here.

We thank the reviewer for this suggestion.

In Anjanappa et al. Nature Communications. 2020, the authors present data which support two main conclusions. First, using X-ray crystallography, the authors suggest that peptide binding to the HLA-A*02:01 groove results in concerted conformational changes of residues at the A- and F-pockets of the binding groove. Second, using molecular dynamics simulations, the authors suggest that binding of the C-terminal peptide residues into the F-pocket propagates a conformational change to the A-pocket resulting in full capture of the peptide.

Our NMR experiments suggest that, instead of participating in a direct, stable interaction with the peptide or floor of the MHC-I groove, the TAPBPR G24-R36 loop promotes the formation of a peptide intermediate that is partially bound at the F-pocket of the MHC-I groove. Putative models involve the TAPBPR G24-R36 loop acting by either reducing the free energy barrier for peptide capture, or by reducing the dissociation of a partially bound “encounter complex” . This enhanced peptide interaction, promoted by the TAPBPR loop, may result in conformational changes at the F-pocket propagating to A-pocket residues to ultimately enable full capture of the peptide within the MHC-I groove.

We have included discussion in **lines 608-619 of the revised manuscript.**

The constraints (LINCS and SETTLE) used in the MD simulations correspond to a 2-fs integrator, but the authors used 4 fs. Probably, they forgot to mention that they also use virtual sites for hydrogens to allow for 4 fs. Otherwise, their simulations would presumably not have been stable.

The reviewer is correct and we apologize for omitting this information in the original version of the manuscript. Indeed, we used virtual site hydrogens (as per Feenstra et al. J Comput Chem 20. 1999) to allow for a 4-fs integration time step. We have included this information in the revised Methods of the manuscript in **lines 847-849.**

Reviewer #3 (Remarks to the Author):

This is a strong and compelling manuscript that addresses the mechanism of tapasin/TAPBPR in peptide loading and exchange. In exquisite detail, the authors have addressed the mechanism of peptide selection and addressed complexities such as the role of peptide properties and the impact of MHC variants. The manuscript is a physical tour de force and provides an excellent example of using fundamental protein biophysics to address challenging immunological questions.

We thank the Reviewer #3 for their positive appraisal of our work, and constructive advice on our ITC experiments. Below we provide a point-by-point response to comments raised.

I have one concern that need to be addressed. The ITC and FP experiments purport to get at binding of peptide to empty MHC-I proteins. To get at this the authors stabilize "empties" using substoichiometric TAPBPR. Stoichiometry notwithstanding, their measurements will still be influenced by the enthalpy of TAPBPR dissociation during the titration. While throughout the text they are careful to use the phrase "apparent K_d ", and they have some language elsewhere that alludes to the fact they are not exactly measuring peptide binding to empties, this point is lost in presentation of Fig. 5A and the discussion around it. Given the rigor elsewhere in the paper, the authors should be more clear about this. While the values obtained are realistic as the authors point out, increasing clarity is important and will not decrease the value or conclusions of the paper.

We agree with the reviewer regarding the contribution(s) of TAPBPR dissociation in our ITC and FP measurements. While the TAPBPR concentration (~50 nM) used in FP experiments for determination of $IC_{50,2}$ corresponds to substoichiometric TAPBPR conditions, it is incorrect to refer to the ITC conditions for K_{D2} as substoichiometric because the concentration of MHC-I/TAPBPR complex used (~15 μ M) is well above the estimated dissociation constant (~190 nM) of the MHC-I/TAPBPR complex, previously determined using SPR experiments on an analogous system (Jiang et al. Science. 2017).

Prompted by reviewer's comments, we have updated Figure 5, Table 1 and the corresponding supplemental figures to specifically denote measured K_{D2} and $IC_{50,2}$ values as "apparent" (we now use $K_{D2,app}$ and $IC_{50,2,app}$). Because we determine K_{D1} from thermodynamic balance in a manner that is dependent on $K_{D2,app}$, we also refer to it as $K_{D1,app}$. We have carefully corrected the use of terms throughout the revised manuscript and explicitly defined the use of the "substoichiometric" in lines 401-404.

Similarly, the authors do not indicate the n values from the ITC experiments. These are necessary to evaluate the quality of the K_d values determined by ITC data, as well as the models chosen. For K_{d2} , it could possibly speak to the impact of TAPBPR dissociation in contributing to heats. As a final point, the concentration of substoichiometric TAPBPR would decrease as sample is injected, further complicating exactly what is measured - this should be clarified, and possibly could be addressed via the n values as indicated above.

We agree with the caveats in interpreting the ITC data brought up by the reviewer, and we apologize for not making this point sufficiently clear in our original text. For measurement of K_{D3} and K_{D4} we obtain stoichiometry (n) values of 1, corresponding to the expected 1:1 stoichiometry of binding (Table 1). We observe similar n values for all peptides analyzed using TAPBPR^{WT} or TAPBPR^{ΔG24-R36}. In contrast, in the ITC experiments measuring $K_{D2, app}$, the resulting n value is 0.4 to 0.5 (Table 1), suggesting that only a portion of HLA-A*02:01 is released from TAPBPR and available for binding to the titrated peptide. The remaining HLA-A*02:01 remains tightly associated with TAPBPR does not bind to peptide under the concentrations used in the ITC experiment. Another important caveat is that the heat from any dissociation of TAPBPR from HLA-A*02:01 could contribute by decreasing or increasing the magnitude of the measured enthalpy. This effect will depend on the concentration of HLA-A*02:01/TAPBPR complex in the ITC cell, further complicating the analysis of our results. Finally, a slow off-rate of TAPBPR from HLA-A*02:01 may also contribute to limiting the concentration of MHC-I available for peptide binding. For these reasons, the ITC determined $K_{D2, app}$ represents an *upper limit* of the dissociation constant and, likewise, the n value is an *apparent* stoichiometry. We note that even though $K_{D2, app}$ reflects an upper limit, a comparison of $K_{D2, app}$ extracted from experiments using different TAPBPR constructs shows very similar values, within experimental error (Table 1). Given that peptide binding to empty MHC-I is independent of TAPBPR, our ITC data suggest no effect of the TAPBPR G24-R36 loop on binding to empty HLA-A*02:01, consistently with our NMR data (Fig. 4C

We have included all n values obtained by ITC in the revised manuscript in the updated Table 1. We have highlighted these important caveats and technical points in the revised manuscript in lines 558-575.

A minor point: what was the rationale for the single mutants that were selected in the targeted mutations in the G24-R36 loop (lines 169-170 on p. 4)? Although the more comprehensive mutational experiments follow, there should be an explanation of why these particular mutations were chosen.

The deep mutational scan does not examine deletions, only single amino acid substitutions. To investigate the loop's function, we began by asking what happens if it is deleted, which necessitated testing mutants individually by targeted mutagenesis. The deletion mutants indicated the loop tip can be deleted without any loss of activity, but there is then a sudden loss of activity when the deletion is extended to L18. We then followed up on this finding by doing targeted substitutions of the loop tip (i.e. G15) and L18, showing only mutations to L18 are deleterious. The rationale behind the choice of these particular mutations was to be highly disruptive, hence the small residue glycine-15 was mutated to the long aliphatic leucine or the charged residue glutamate, while hydrophobic leucine-18 was mutated to small glycine or long charged residues glutamate and lysine. We have clarified this final point in the text (lines 180-182):

“Targeted substitutions of tapasin L18, chosen to be highly disruptive based on altered side chain properties, caused similar decreases in activity, whereas substitutions of G15 at the very tip of the loop had minimal effect.”

REVIEWER COMMENTS

Reviewer #1 (Remarks to the Author):

The authors have addressed all my concerns.

Reviewer #2 (Remarks to the Author):

The authors have adequately addressed all the concerns I raised in my review. I recommend accepting the revised manuscript for publication.

--

Olivier Fiset, Ph.D.
Advanced Research Computing, ICT
University of Saskatchewan, Canada

Reviewer #3 (Remarks to the Author):

The authors have addressed my concerns, and from what I could tell done a great job responding to those of the other reviewers.

Reviewer #1 (Remarks to the Author):

The authors have addressed all my concerns.

We appreciate Reviewer #1 for their constructive feedback.

Reviewer #2 (Remarks to the Author):

The authors have adequately addressed all the concerns I raised in my review. I recommend accepting the revised manuscript for publication.

--

Olivier Fiset, Ph.D.
Advanced Research Computing, ICT
University of Saskatchewan, Canada

We thank Reviewer #2 for their helpful suggestions and comments.

Reviewer #3 (Remarks to the Author):

The authors have addressed my concerns, and from what I could tell done a great job responding to those of the other reviewers.

We are grateful to Reviewer #3 for their constructive feedback and appraisal of our work.

A sincere thank you to all reviewers.

Their comments have helped us improve our manuscript considerably.